# Unexpected behaviors in molecular transport through size-controlled nanochannels down to the ultra-nanoscale

Giacomo Bruno[1,2], Nicola Di Trani[1,2], R. Lyle Hood[1], Erika Zabre[1], Carly Sue Filgueira [1], Giancarlo Canavese[3], Priya Jain[1], Zachary Smith[1], Danilo Demarchi[2], Sharath Hosali[4], Alberto Pimpinelli[1,5,6], Mauro Ferrari[1] & Alessandro Grattoni[1]

Ionic transport through nanofluidic systems is a problem of fundamental interest in transport physics and has broad relevance in desalination, fuel cells, batteries, filtration, and drug delivery. When the dimension of the fluidic system approaches the size of molecules in solution, fluid properties are not homogeneous and a departure in behavior is observed with respect to continuum-based theories. Here we present a systematic study of the transport of charged and neutral small molecules in an ideal nanofluidic platform with precise channels from the sub-microscale to the ultra-nanoscale (<5 nm). Surprisingly, we find that diffusive transport of nano-confined neutral molecules matches that of charged molecules, as though the former carry an effective charge. Further, approaching the ultra-nanoscale molecular diffusivities suddenly drop by up to an order of magnitude for all molecules, irrespective of their electric charge. New theoretical investigations will be required to shed light onto these intriguing results.

[1] Department of Nanomedicine, Houston Methodist Research Institute, Houston, TX 77030, USA. [2] Department of Electronics and Telecommunications, Politecnico di Torino, 10024 Turin, Italy. [3] Department of Applied Science and Technology, Politecnico di Torino, 10024 Turin, Italy. [4] NanoMedical Systems, Inc., Austin, TX 78744, USA. [5] Smalley-Curl Institute, Rice University, Houston, TX 77005, USA. [6] Department of Material Science and Nanoengineering, Rice University, Houston, Texas 77005, USA. Correspondence and requests for materials should be addressed to A.G. (email: agrattoni@houstonmethodist.org)

Physiological regulation of molecular transport in complex organisms is precisely controlled at picomolar specificity[1]. For instance, the passage of organic anions and cations, as well as drugs and xenobiotics, are regulated at the molecular level by transmembrane solute carriers to maintain homeostasis[2]. Intracellular signaling and stimuli, as well as cell-to-cell and cell-to-matrix physiological transport, can be effectively mimicked by nanofluidic systems[3,4]. Such systems are also currently utilized in multiple fields, including desalination[5,6], cell and drug delivery[7–9], analyte sorting[10], and filtration[11], among others. Unique transport behaviors can be attained when diffusion occurs under nanoscale spatial confinement, especially when the size of the fluidic systems approaches the size of fluid molecules. At this scale, the media is no longer homogeneous, fluctuations in density and viscosity have been predicted[12]. Unfortunately, investigations of transport at this scale have been limited by unsurmountable technological challenges in generating reliable membranes presenting a defined number of channels with tightly controlled shape, geometry, and sizes. Studies leveraging microfabricated structures have suffered from the difficulty to fabricate nanofluidic systems with a high number of channels. As such, most investigations have relied on experimental results obtained with few channels and minute outputs[13,14]. More recent studies using carbon nanotubes, alumina, silicon, or titania nanoporous films have similarly suffered from widely fluctuating channel and pore dimensions, leading to significant uncertainties in the theoretical interpretation of results[15–17]. It is not by chance that studies investigating electrostatic effects on transport under confinement[18,19] have relied on relatively large nanochannels and on the modification of the ionic strength of the solution rather than on the reduction of the channel size. This approach limited the investigation to the case in which fluids still behave as continuum. In contrast, we employed robust and scalable nanofluidic membranes to investigate the concentration-driven transport of molecules within physical spaces at the lower end of the nanoscale, experimentally accounting for the complexity of fluid at this scale. In this context, we performed a systematic experimental investigation using a scaled series of nanochannels with discrete, monodispersed sizes (2.5–250 nm).

In this paper, we present previously unreported diffusive behaviors of molecules down to the ultra-nanoscale (<5 nm); when compared with existing models, our results suggest that present-day understanding of transport of charged and neutral molecular species in nanochannels is limited. Also, at the ultra-nanoscale, concentration-driven diffusion was observed to exhibit a linear release profile regardless of molecular charge. A fundamental parameter in electrostatic contributions with highly spatially confined nanofluidic channels is the electrical double layer (EDL), which has a characteristic dimension known as Debye length[20]. A Debye length greater than one-tenth the nanochannel height leads to appreciably overlapping EDL, as estimated by the Debye–Hückel approximation. High spatial and electrostatic confinement can be combined to achieve constant solute transport. The width of the EDL is inversely proportional to ionic concentration, as increased ion screening reduces Debye length. Under physiological conditions, ion concentration gradients and electric forces drive the flow of ions through channels, processes which remain essential for cell function and survival. In general, nanoconfined diffusion for charged molecular species can be understood in the framework of the Poisson–Boltzmann equation, which leads to the concept of the EDL[21]. This description is expected to be insignificant for neutral solutes, whose transport should be regulated by hard-sphere (HS) and hydrodynamic interactions with channel walls, when the size of channels approaches the size of the molecules.

Here, we present the observation of an unexpected transport regime where the diffusion flux appears to be determined by the electrostatic potential of the electrolyte solution, irrespective of the charged or neutral state of the diffusing molecules. In this regime, all molecules appear to possess an electric charge, either actual or effective. Importantly, the observed sudden decrease of the effective diffusivity of all cationic, anionic, and neutral species at the ultra-nanoscale could not be justified within the framework of existing models exclusively based on electrostatic, HS, and hydrodynamics interactions. Other factors, currently unaccounted for, must be at play. Insights from this study expand our experimental knowledge of nanoconfined molecule dynamics, which is relevant for biological systems as well as for technologies ranging from synthetic membranes to batteries.

## Results

**Nanochannel membrane design and characterization.** The silicon membranes utilized in this study were manufactured through a complex sacrificial etching process[22]. Nanochannels were obtained by hydrogen peroxide etching of physical vapor deposited tungsten. The resulting 340,252 slit-channels per membrane presented a highly defined, precise geometry with a few Å size tolerance and were obtained parallel to the membrane surface and orthogonal to inlet and outlet microchannels. This design promotes high channel density and mechanical robustness (Fig. 1a). Scanning and transmission electron microscopy (SEM and TEM) images (Fig. 1c–i) confirmed successful nanofabrication for nanochannels heights of 2.5, 3.6, 5.7, 13, 20, 40, and 250 nm. Figure 1j and k presents gas flow and selective quantum dot filtration highlighting accurate nanochannel manufacturing at the population level. The pressure-driven nitrogen flow results tightly correlated with kinetic gas flow predictions. Neutral charge quantum dot filtration in toluene exhibited precise selectivity of our nanofluidic architecture with a pass/no-pass resolution of 0.5 nm. These results provide strong confidence in the effective nanochannel size and transport regulation (see Supplementary Notes 1–4). This unique nanofluidic platform allowed for a broad comparative analysis of transport properties between differently charged analytes, through channels ranging from ultra-nanoscale to the sub-microscale.

**Assessing diffusive transport in nanochannels.** By using UV spectrophotometry, we measured the cumulative release of six different molecules (two positive, two negative, and two neutral, whose physicochemical properties are summarized in Table 1) as a function of time, through different membranes, each one characterized by nanochannels of one given height $h$. All the release experiments were performed in a 50 mM NaCl solution (corresponding to a Debye length ~2 nm). The nominal solution pH for each experiment (see Table 1) was obtained by adding small quantities of HCl or KOH. Figure 2 shows measurements of cumulative mass release for three of the molecules, representative of positive, negative, and neutral solutes, respectively.

**Positively charged molecules.** The transport behavior for positive analytes (histamine in Fig. 2a; epinephrine in Supplementary Fig. 5a) separates into two distinct regions by nanochannel size. The first region (5.7–250 nm for histamine) shows cumulative mass transport curves that follow a $1-\exp(-t/\tau)$ profile, where $t$ is time and $\tau$ is the diffusion timescale. The release plateau reached for the largest nanochannels is due to the depletion of molecules from the source reservoir of the release test system[23]. The cumulative release from channels with $h = 5.7$ up to 40 nm appears to be independent of $h$. This may seem surprising at first, but is in fact consistent with the previously theorized "near-surface diffusion" behavior[24,25], which arises from near-surface cationic localization along negatively charged walls of the silicon

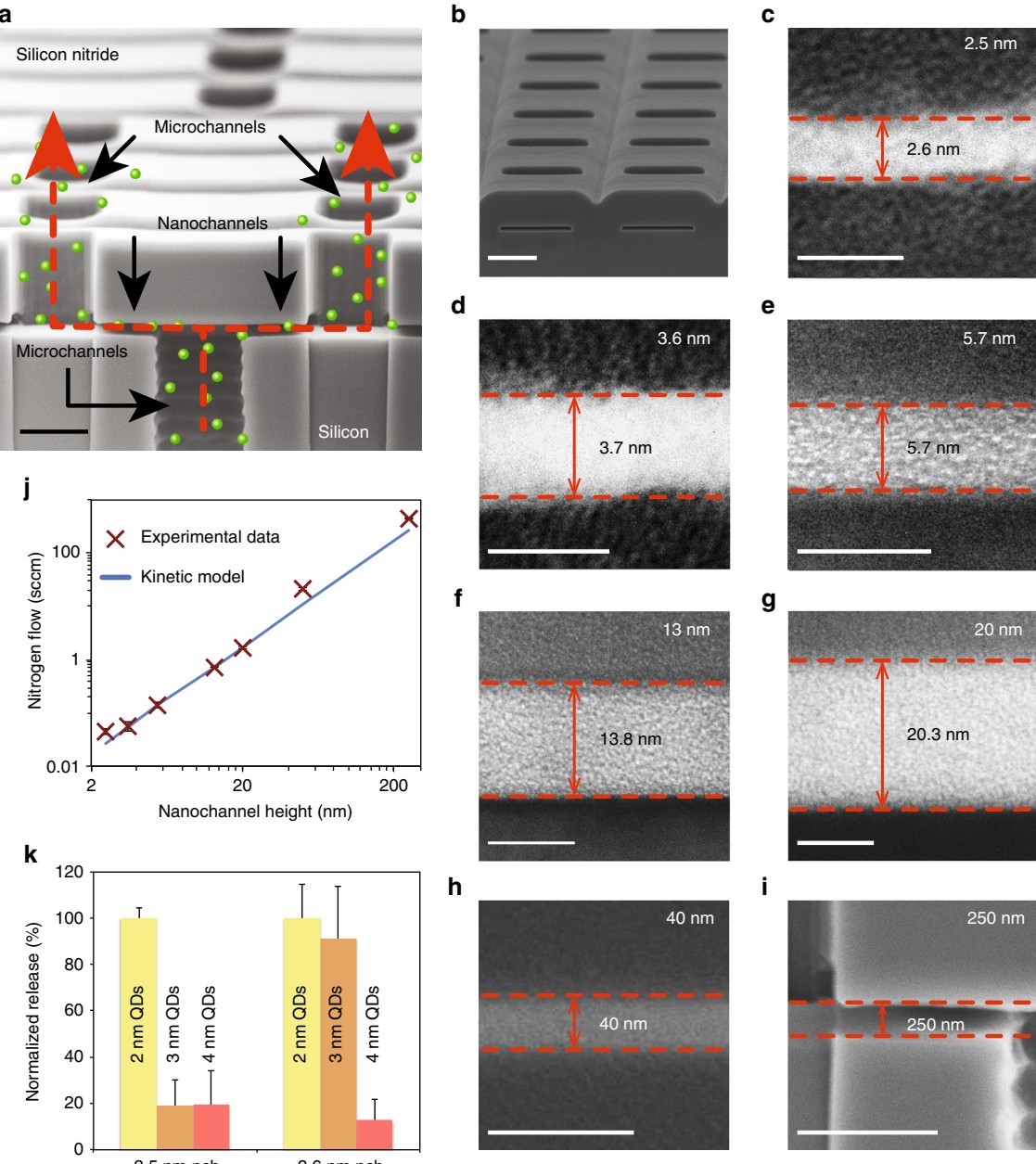

**Fig. 1** Characterization of the nanochannel membranes used in this study. **a** SEM cross-section of the silicon membrane structure illustrating the molecular pathway through the channels (the particles in green represent molecules not to scale). The inlet and outlet microchannels present a width of 3 µm, a height of 1 µm, and a respective length of 15 and 1.7 µm. The nanochannels present a width of 3 µm, a length of 1 µm, and height of 2.5–250 nm. Scale bar: 1 µm. **b** Lateral view SEM image of a sectioned nanochannel membrane. Scale bar: 2 µm. **c–h** TEM images of 2.5 to 40 nm nanochannel cross-sections. Scale bars: 5 nm (**c, d**), 10 nm (**e–g**), and 100 nm (**h**). **i** SEM image of 250 nm nanochannel cross-section. Scale bar: 1 µm. **j** Gas testing results of transmembrane nitrogen ($N_2$) flow (symbols) showed strong correlation to kinetic model predictions (solid line). These analyses allow for the evaluation of the entire nanochannel population, as opposed to single-channel assessment through SEM/TEM. Error bars are within the marker size for experimental data and the nanochannel sizes maintained a maximum percentage standard deviation of 15%. **k** Selective filtering of 2, 3, and 4 nm (±0.5 nm) neutral quantum dots (QDs) by 2.5 and 3.6 nm nanochannels (nch). Data exhibited tight size selectivity with a resolution of 0.5 nm. Release is normalized to the fluorescence intensity of the 2 nm QDs. Each of the data for both **j** and **k** denotes the average of three individual replicates, error bars are ±sd

slit-nanochannels, when the distance $h$ between the walls becomes of the order of 10 times the Debye length. Redistribution of the diffusing molecules within an electrostatically defined space adjacent to the wall surface is expected to result in transport governed by surface area rather than channel volume, and therefore independent of $h$. For histamine (Fig. 2a) inside the nanochannels whose heights range from 5.7 to 40 nm, near-surface diffusion would explain why an increase in channel cross-

section of more than 700% leads to insignificant differences in cumulative release. As nanochannels have a length $L$ and width $w$ of 1 and 3 µm, respectively, their surface area $2wL(1 + h/w)$ does not vary significantly until the contribution from the side walls, and eventually from bulk diffusion becomes relevant at the largest (250 nm) channels.

In the second region ($h < 5.7$ nm for histamine, $h < 13$ for epinephrine), which is characterized by a zero-order release

**Table 1 Molecular properties of analytes tested**

| Molecule | Histamine | Epinephrine | Aspirin | Phenylalanine | Cefazolin | 3-Aminosalicylic acid |
|---|---|---|---|---|---|---|
| Charge ($q$) | 2 | 1 | 0 | 0 | −1 | −1 |
| Solution pH | 4 | 7 | 3 | 7 | 7 | 7 |
| Radius (Å) | 3.9 | 4.9 | 4.8 | 4.6 | 6.5 | 4.4 |
| Weight (Dalton) | 111 | 183 | 180 | 165 | 454 | 153 |
| LogD | −3.5 | −2 | 1 | −0.4 | −4.5 | −1.5 |
| Concentration | 30 mg ml$^{-1}$ | 150 µg ml$^{-1}$ | 1 mg ml$^{-1}$ | 25 mg ml$^{-1}$ | 2 mg ml$^{-1}$ | 700 µg ml$^{-1}$ |
| Polarizability (Å$^3$) | 12.27 | 19.21 | 17.89 | 17.15 | 39.86 | 14.46 |
| Effective diffusivity at 250 nm (cm$^2$ s$^{-1}$) | $1.4 \pm 0.1 \times 10^{-6}$ | $8.6 \pm 1 \times 10^{-8}$ | $2.3 \pm 0.1 \times 10^{-7}$ | $3.1 \pm 0.3 \times 10^{-6}$ | $1.4 \pm 0.3 \times 10^{-6}$ | $9.4 \pm 0.5 \times 10^{-7}$ |

Physical properties computed with the online chemical calculator[48]. LogD is a measure of hydrophobicity of molecules inclusive of the ionized and non-ionized species; a positive value means hydrophobic character. The molecular radius was computed as the average between the maximum and the minimum projected radius of the molecule, as obtained from ref. [48]

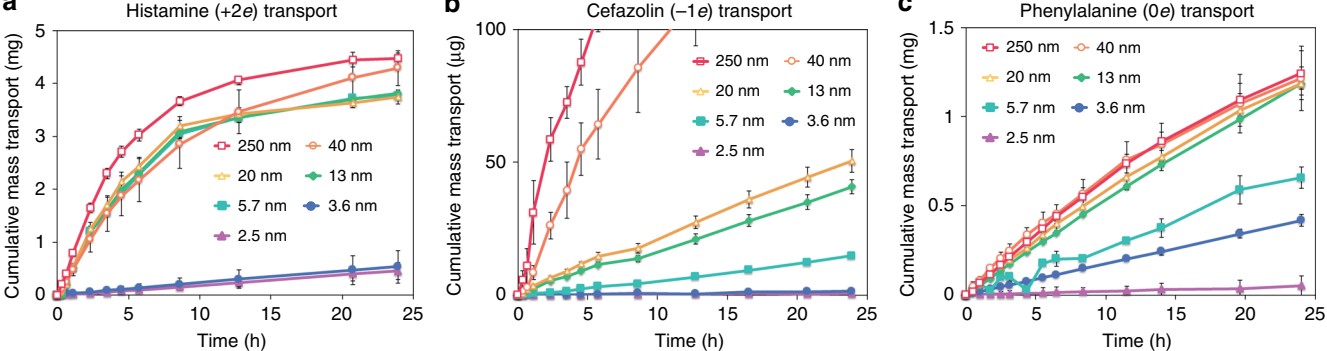

**Fig. 2** Drug transport through nanochannels. Cumulative diffusive mass transport of histamine (positive) (**a**), cefazolin (negative) (**b**), and phenylalanine (neutral) (**c**), chosen as representative of the three different charge states. Release curves are measured for a scaled series of nanochannels over 24 h. The released molecules were dissolved in a 50 mM NaCl aqueous solution within a dual reservoir system housing source and drain chambers separated by a nanochannel membrane. The concentration of molecule released from the source into the sink reservoir was measured via UV spectrophotometry. Each of the data points denotes the average of three individual replicates, error bars are ±sd

profile (constant release rate), a sharp and unexpected decrease of the release rate was observed between 5.7 and 3.6 nm channels for histamine and 13 and 5.7 nm for epinephrine (see Fig. 3 and Supplementary Fig. 5a). It is important to note that obtaining a linear release profile for a molecule carrying a charge of sign opposite to that of the channel surfaces is trivial: as a matter of fact, it is a previously unreported achievement.

**Negatively charged molecules**. Figure 2b exhibits release curves of anionic cefazolin, (3-aminosalicylic acid in Supplementary Fig. 5b) offering a clear depiction of "gated diffusion", where repulsion of negative molecules by the negatively charged silica walls provides electrostatic hindrance to transport. Substantial differences with respect to the cationic release profiles are evident, as release rates increased significantly with minor increments in nanochannel height. For the 40 and 250 nm channels, a pure 1 −exp(−$t/\tau$) profile was observed, which is typical of microfluidic bulk diffusion, suggesting insufficient spatial and electrostatic confinement to linearize transport. Release through the 5.7–20 nm channels was linear for the duration studied, which has been demonstrated previously[24,26,27]. Nonetheless, transport within the ultra-nanoscale, 2.5 and 3.6 nm channels reproduced the substantial decrease in release rate seen for positive analytes (the corresponding behavior observed with anionic 3-aminosalicylic acid is shown in Supplementary Fig. 5b).

**Neutral molecules**. Transmembrane transport of the neutral analytes phenylalanine (Fig. 2c) and aspirin (Supplementary

Fig. 5c) exhibited unexpected features. Both molecules maintained far more linear profiles than the cations (Fig. 2a) at channel heights of 5.7 nm and above (first region); however, aspirin appears to behave similarly to histamine, in that its cumulative release is only weakly dependent on channel size. On the other hand, phenylalanine closely mimics the behavior of anions. However, the release through the 2.5 and 3.6 nm channels exhibited a strong decrease in transport rates, similar to that of charged molecules. As the spatial distribution of neutral analytes, and therefore their release rate, should in principle be unaffected by the charges along the channel walls and in solution, the similarities that the release of aspirin and phenylalanine share with cations and anions, respectively, are extremely surprising and intriguing.

**Effective diffusivity**. To shed light on this diverse set of transport profiles we extracted the effective diffusivity of the analytes in the nanochannels based on the experimental release curves. We fitted the cumulative release curves to the expression $M(t) = V_1 V_2/(V_1 + V_2) \Delta C_0 [1 - \exp(-t/\tau)]$, which gave us the diffusion timescale $\tau$. $V_1$ and $V_2$ are the volumes of the source and sink reservoir, respectively, and $\Delta C_0$ their difference in concentration at $t = 0$. From the diffusion timescale, it is possible to calculate the membrane permeability $P_{mem} = V_1 V_2/(V_1 + V_2)\ 1/\tau$. To obtain the effective diffusivity inside the nanochannels, we accounted for the fact that they are connected in series with inlet and outlet microchannels. Similarly to bulk values, the diffusivities of tested molecules in microchannels ($D_\mu$) are not available in the

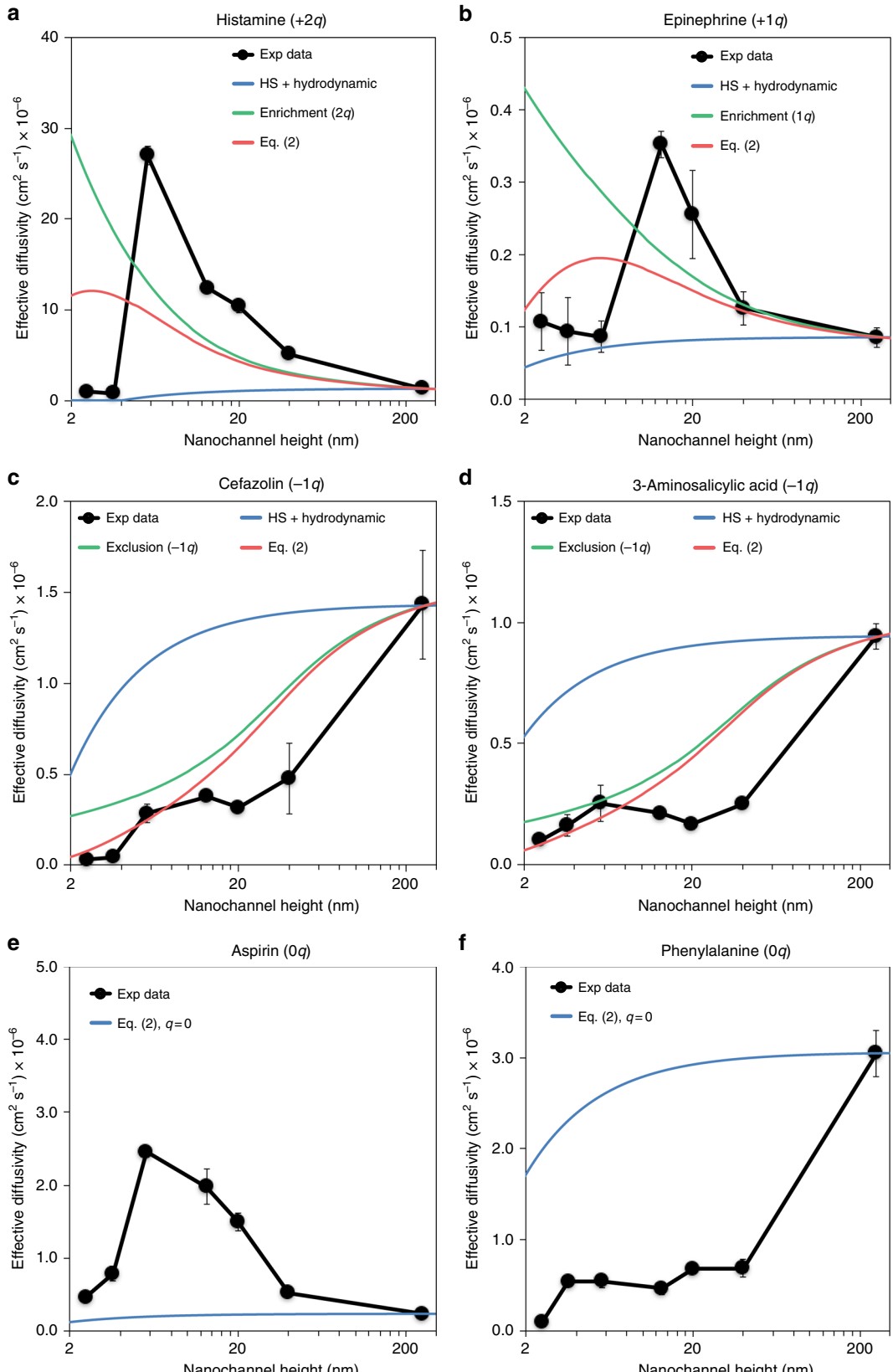

**Fig. 3** Effective diffusivity calculated for each analyte and corresponding theoretical prediction based on Eq. (2). Effective diffusivity for positive (**a**, **b**), negative (**c**, **d**), and neutral (**e**, **f**) molecules. The experimental data (black solid circles) are compared in **a**–**d** with Eq. (2) including only hard-sphere (HS) and hydrodynamics effects (blue solid lines), only exclusion or enrichment effects (green solid lines), and the full Eq. (2) (red solid line). The green line is computed from Eq. (2) with $r_s = 0$ and $K = 1$, corresponding to neglecting HS and hydrodynamic interactions. **e**, **f** show diffusivity data for two neutral species. As in the previous panels, blue solid lines are computed from Eq. (2) with $q = 0$. Each of the data points denotes the average of three individual replicates, error bars are ±sd

literature; their accurate experimental measurement would be highly not trivial. However, the diffusivity of the solutes in microchannels can be expected to be close to the effective diffusivity within the largest nanochannel, with $h = 250$ nm (Table 1) so that assuming $D_\mu = D_{250nm}$ we can write

$$P_{mem} = \left( \frac{L_i}{w_i h_i N_i D_{250nm}} + \frac{L_n}{w_n h_n N_n D_n} + \frac{L_o}{w_o h_o N_o D_{250nm}} \right)^{-1},$$

(1)

where the indices "i", "n", and "o" refer to "inlet", "nanochannels", and "outlet", respectively, and $N_i$, $N_n$, and $N_o$ are the number of microchannels or nanochannels in the three respective regions. From Eq. (1) we obtain $D_n = D_n(h)$ for each channel height $h$. Results are shown in Fig. 3 for all of the solutes.

**Effective diffusivity of positive molecules.** The effective diffusivities of histamine (black solid circles in Fig. 3a) and epinephrine (black solid circles in Fig. 3b) have the same qualitative behavior: in large channels ($h > 40$ nm) the diffusivities are approximately constant, and, we expect the values in the largest, $h = 250$ nm channels to be close to the value of the diffusivity in the micrometrical sections of the membranes. For smaller channels, the effective diffusivities $D_n(h)$ are seen to increase with decreasing $h$. This implies that the solute distribution inside the nanochannel becomes strongly non-uniform: positive molecules tend to concentrate close to the channel walls, where the negative potential becomes larger in magnitude, so that the diffusion flux from the bulk of the channel becomes negligible. This is an example of a near-surface diffusive regime. Finally, at the ultra-nanoscale the diffusivities drop abruptly by more than an order of magnitude.

**Effective diffusivity of negative molecules.** The diffusivities of negatively charged cefazolin and 3-aminosalicylic acid (black solid circles in Fig. 3c, d) are seen to monotonically decrease from their value in $h = 250$ nm channels, then to remain approximately constant when $5.7$ nm $< h < 40$ nm, and eventually to drop again for channels with $h < 5.7$ nm. While the initial decrease of the diffusivities can be expected as a result of electrostatic exclusion of anions from the channels, the plateau for $5.7$ nm $< h < 40$ nm, as well as the further drop of diffusivity for $h < 5.7$ nm, are puzzling.

**Effective diffusivity of neutral molecules.** Two aspects of the experimental results for aspirin and phenylalanine (black solid circles in Fig. 3e, f) are noteworthy. First, the diffusivities of these neutral molecular species differ completely from one another. Second, and even more strikingly: on the one hand the $h$-dependence of the diffusivity of aspirin (Fig. 3e) is indistinguishable from that of positively charged species, especially histamine in Fig. 3a; on the other hand, the diffusivity of phenylalanine closely matches the diffusivity of the negatively charged molecules of Fig. 3c and d. The fact that different neutral molecules can exhibit different diffusion behavior from one another is in itself surprising, though it becomes less so when one realizes that aspirin is a strongly hydrophobic molecule, which is what the positive value of logD in Table 1 implies. As such, it may be expected to diffuse next to the channel walls, and thus to exhibit a near-surface diffusion behavior mimicking that of positively charged molecules. However, the behavior of phenylalanine is completely unexpected. The uncanny resemblance with the diffusivity of cefazolin, and especially of aminosalicylic acid, appears to suggest that neutral molecules may possess in certain situations an "effective charge", whose origin is, however, a mystery.

**Model calculations.** The standard approach to modeling the diffusivity of charged or neutral molecules inside slit-channels of variable heights, containing a NaCl solution at equilibrium between two reservoirs, consists in several steps: first of all, computing the electrostatic potential $\psi(z)$ across the channel height, due to a 1–1 electrolyte (NaCl) from the Poisson–Boltzmann equation; then, computing the equilibrium distribution of solute molecules in the potential $\psi(z)$; finally, computing the effective diffusivity of the solute as $D_{eff} = \beta D_{bulk}$, where the so-called partition coefficient $\beta$ depends on $\psi(z)$ (unless the solute is neutral) and on the HS and hydrodynamic interactions of the dissolved molecules with the channel walls. While we summarize the model here, additional details are provided in Supplementary Notes 9–11.

Knowing the electrostatic potential inside the channels requires knowledge of the charge at the surface of the channel walls. Following Behrens et al.[28], the surface charge on the silica channel walls was self-consistently computed using a site-binding model for the silica protonation-deprotonation reaction, at equilibrium with the total charge in the solution. Importantly, the model accounts for the decrease in pH inside the nanochannel which is expected at decreasing channel height. Two competing processes are balanced: (i) the increasing attraction of positive ions into the channel due to the negatively charged silicon surfaces as the surface-to-volume ratio increases; (ii) the decrease in surface charge density of the silicon surfaces due to the decrease of the solution pH. Both effects are iteratively adjusted until convergence is achieved. As a result, the pH in nanochannels with $h = 2.5$ nm is predicted to be approximately equal to 6.2, when the bulk solution is kept at a pH equal to 7.

To validate the site-binding model for our membranes, as well as the choice of its parameter values—including surface sites density ($N_s = 1.5$ sites nm$^{-2}$), equilibrium constant (pKn = 7.5), capacity of the silica–water interface ($C = 2.9$ C m$^{-2}$), we have measured the surface potential as a function of nanochannel size, with the method of the streaming potential (see details in Supplementary Note 16). Good qualitative, and semi-quantitative, agreement was found between predictions of the site-binding model (red solid line in Fig. 4) and experimental data (black solid dot in Fig. 4). In particular, the position of the maximum of the surface potential is correctly captured.

Once the electrostatic potential of the solution, $\psi(z)$, was known, we computed the partition coefficient $\beta$ as follows:

$$\beta = \frac{1}{h} \int_0^h K^{-1} \exp\left( -E_{HS} - q\frac{\psi(z)}{k_B T} \right) dz = \frac{1}{h} \int_{r_s}^{h-r_s} K^{-1} \exp\left( -q\frac{\psi(z)}{k_B T} \right) dz,$$

(2)

where the solute concentration in the channel is assumed to follow from the Boltzmann distribution $\exp[-(E_{HS} + q\psi(z))/ k_B T]$. Here, $E_{HS}$ is a HS potential confining the particles inside the channel, $q = Ze$ the electric charge of the solute, $r_s$ is the solute molecule's radius, $z$ is the distance from the channel surface, $k_B$ and $T$ are the Boltzmann constant and the temperature, respectively. Following Ganatos et al.[29], hydrodynamic interactions were accounted for through a drag coefficient with the centerline approximation, $K^{-1}$, given in Supplementary Note 12.

Obviously, the spatial distribution of neutral particles (for which $q = 0$) is expected to depend only on the HS and hydrodynamic interactions; following Smith and Deen[30,31], the neutral partition coefficient should then read: $\beta_{st+h} = (h - 2r_s)/h \cdot K^{-1}$.

A few words of caution are needed at this point. The molecules studied in this work are far from being spherical, so that the "HS potential" must be understood as translating the reduction in the —mostly rotational—degrees of freedom of the diffusing molecules in narrow channels; the "molecular diameter" is thus an unknown effective quantity. Maybe even more important, the drag

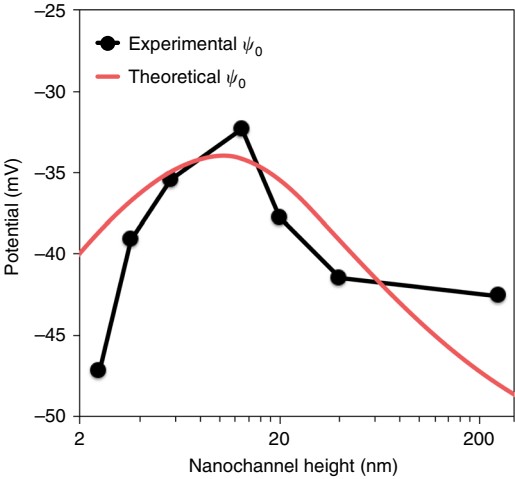

**Fig. 4** Experimental and theoretical surface potentials. Surface potential $\psi_0$ as a function of nanochannel height computed with the site-binding model for the silica surfaces (red line). Experimental measurements of the surface potential obtained through the streaming potential method (black dots) using a 50 mM NaCl solution

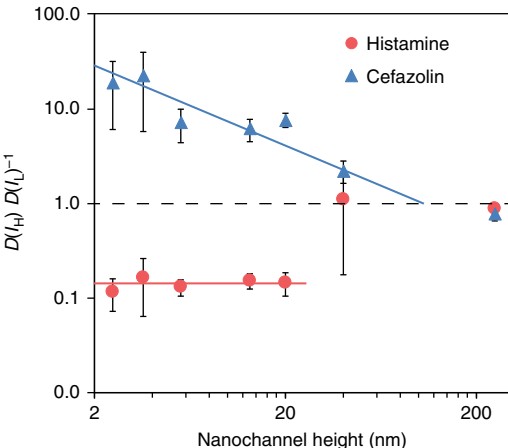

**Fig. 5** Drug diffusion in high and low ionic strengths bulk solutions. Ratio between effective diffusivities obtained for histamine (solid dots) and cefazolin (solid triangles) at high (137 mM NaCl, $I_H$) and low (deionized water, $I_L$) ionic concentration for each nanochannel size. Each of the data points denotes the average of three individual replicates, error bars are ±sd

coefficient is defined for spherical particles much larger than the solvent, and computed assuming that they move only within the central surface of the channel (centerline approximation). Last, but not least, the electrostatic interactions between solute molecules and channel walls are computed with the assumption that introducing a finite concentration of charged molecules in the solution does not change its potential. This assumption, which lies unquestioned at the foundation of every solute diffusion model, is clearly wrong, in principle. However, going beyond this assumption is well beyond the scope of the present paper. These considerations must be kept in mind while evaluating the comparison between experimental data and model calculations.

**Comparison between model and experimental results**. The comparison of model vs. experimental diffusivities is presented in Fig. 3. In panels (a–d), we plotted Eq. (2) as a solid red curve; we also plotted separately the electrostatic contribution (obtained by letting $r_s = 0$ and $K = 1$, shown as a green line) as well as the HS plus hydrodynamic interactions part (obtained by letting $q = 0$, shown as a blue line). In panels (e, f), we plotted only Eq. (2) with $q = 0$. For each solute, we assumed values of molecular diameters $d = 2r_s$ with $r_s$ from Table 1. As mentioned above, our molecules are not spherical, and it is unclear what their effective diameter in solution should be. Equation (2) performs poorly for positive solutes, for which it predicts a maximum of diffusivity in the ultra-nanoscale, where diffusivities are in fact the smallest. The position of the maximum depends on the choice of $r_s$ and might point to a larger value of the effective molecular radius in the solution. However, the height of the maximum depends on the molecular charge, and can be only matched to the experimental data if a much larger charge is assumed on each solute molecule. We discuss these attempts at fitting the data with Eq. (2) using $r_s$ and $q$ as free parameters in the Supplementary Note 15, though we stress that such fits have, at the moment, no physical interpretation. One should notice that a similar strong increase of the experimental diffusivity with decreasing $h$ was also observed by Plecis et al.[19] who ascribed it to the cationic enrichment effect described by the green curve in Fig. 3a and b. However, they neglected in their description the HS and hydrodynamic interactions that we added here.

**Positive molecules**. Effective diffusivity results obtained for histamine at high (137 mM NaCl) and low ionic strength (deionized, DI water) support the role of enrichment on the transport of cations in nanochannels. Figure 5 shows the ratio between effective diffusivities obtained for histamine at high and low ionic concentration for each nanochannel size (solid red dots).

Two important observations can be made: first, the increase in ionic strength determines a reduction in effective diffusivity consistently for all nanochannel sizes below 20 nm. This implies that the near-surface diffusion of histamine does not depend on the ratio $\lambda_D/h$. Second, this provides evidence that enrichment effects cannot be responsible for the drop in effective diffusivity at the ultra-nanoscale.

In fact, no significant differences are observed for larger nanochannel sizes. In contrast, one order of magnitude higher diffusivity is obtained in DI water as compared to respect to high ionic concentration for 2.5 and 3.6 nm channels. While the enrichment model seems to explain the relative difference in diffusivity at different ionic strength, it fails to capture the abrupt decrease in the diffusivity at the ultra-nanoscale ($h < 5.7$ nm). This effect, which is more pronounced for histamine than for epinephrine, is not even explained by considering steric and hydrodynamic interactions (red solid curve). By fitting the data with the model accounting for enrichment, steric, and hydrodynamic effects, and using the charge and the radius of the solute as free parameters a better qualitative fit for both positive solutes can be achieved (see red dash lines in the Supplementary Fig. 9). However, the increase in the molecular charge obtained is difficult to justify. It is apparent, therefore, that mean-field models like the present one are not able to capture the subtleties of the ultra-nanoscale, where fluctuations in fluid properties are expected to become dominant[32].

**Negative molecules**. The model Eq. (2) appears to be qualitatively correct in predicting that the diffusivity of negative molecular solutes is monotonically decreasing (Fig. 3c, d). Physically, this is a consequence of anion exclusion from a negatively charged nanochannel. The agreement between the model and the measured diffusivity does not go beyond qualitative, though. In particular, the model fails to capture the diffusivity plateau at intermediate length scales (5.7 nm < $h$ < 40 nm) as well as the sudden drop in

diffusivity at the ultra-nanoscale. However, effective diffusivity results obtained at high and low ionic strength (Fig. 5, blue solid triangles) confirm the role of exclusion on the transport of anions in negatively charged nanochannels. As expected, an increase in effective diffusivity at higher ionic strength is observed for nearly all nanochannel sizes. This increase becomes consistently smaller at increasing nanochannel size, proportionally to a power of $\lambda_D/h$, until it vanishes for 250 nm. Although clearly relevant in the transport of negative charges in nanochannels, exclusion cannot explain the nearly constant diffusivity in the $h = 5.7$–40 nm length region (Fig. 3c, d) nor the drop in diffusivity at the ultra-nanoscale. If we include HS and hydrodynamic interactions to try to capture the ultra-nanoscale sudden drop of the experimental diffusivities, we further decrease the overall agreement: the charge-independent ultra-nanoscale behavior eludes understanding in this case as well.

**Neutral molecules.** Turning now to aspirin and phenylalanine (Fig. 3e, f) we feel quite comfortable with assuming that these molecular species are indeed neutral in our experimental conditions; for aspirin, we performed the experiments at pH 3 in the bulk solution, so that the hydroxyl group of aspirin (pKa = 3.5) is expected to dissociate minimally leaving aspirin molecules neutral (see Supplementary Fig. 2). Release of phenylalanine was measured at pH 7 in the bulk solution; the molecule has two pKa values, due to the simultaneous presence on it of both a hydroxyl and an amine group. At very acidic pH, the OH group does not dissociate, while the $NH_2$ protonates, so that the molecule presents a positive charge. At pH ~6, as estimated inside the nanochannels, the molecule is expected to be neutral, and to acquires a negative charge only at pH > 9, when the OH group dissociates (see Supplementary Fig. 3).

Neutral solutes, being unaffected by electrostatic, are expected to interact with the channel walls only through repulsive HS and hydrodynamic interactions, and attractive van der Waals (vdW) forces. The model in Eq. (2) neglects the latter, but it is clear from Fig. 3e and f that Eq. (2) with $q = 0$ has little to do with the experimental data, and it is hard to see how any vdW attraction may modify the trends observed for aspirin and phenylalanine. As a matter of fact, and as briefly mentioned earlier, the diffusivity of aspirin has a very different behavior, compared to that of phenylalanine, because aspirin is strongly hydrophobic, as implied by the positive logD value in Table 1. In light of the discussion of the near-surface diffusivity of positive molecules, it can be argued that aspirin tends to segregate along the silica walls, and that this is why its transport mimics the quasi 2-dimensional near-surface diffusion of positively charged histamine and epinephrine. As a matter of fact, hydrophobic interactions are usually modeled as decreasing exponentials, reminiscent of screened electrostatic interactions[33].

It would be therefore not unreasonable to use Eq. (2) to model aspirin's diffusivity, with an "effective positive charge" mimicking an attractive hydrophobic interaction with the silica walls. The model is very crude, but the agreement with the data is acceptable (dashed red line, Supplementary Fig. 9).

The most difficult conundrum is provided by phenylalanine's diffusivity: the latter closely follows the same $h$-dependence as the negatively charged cefazolin and aminosalicylic acid, and even exhibits a very similar plateau of nearly constant values for $h$ between 3.6 and 40 nm, with the omnipresent, charge-independent sudden drop in the smallest nanochannels. The data strongly suggest the tantalizing possibility that phenylalanine molecules were able to interact with the electrostatic potential in the channel, as if they were carrying an "effective negative charge", though it is by no means clear what the origin of this effective charge might be.

## Discussion

All molecules, as shown in Fig. 3, share a very similar charge-independent ultra-nanoscale sudden drop in diffusivity. As it is clear from the figure, HS and hydrodynamic interactions are not able to capture this behavior. At this very small scale, adsorption of diffusing molecules onto the channel walls could lead to a reduced diffusivity. Plecis et al.[19] mention observing a slightly reduced diffusivity of positively charged Rhodamine 6G, but they attribute it to electrostatic adsorption: in fact, it only happens when the channel walls are negatively charged. In our experiments, a quite different scenario should be conceived: at the ultra-nanoscale, electrostatic solute-wall repulsion or attraction would be suddenly negligible, and completely overcome by charge-independent attractive forces, such as vdW interactions. In Table 1, we list for each solute their polarizability per unit volume, which turns out to be very close to 0.12 for all molecules. vdW interactions are known to scale as polarizability divided distance to the power 6. The negatively charged cefazolin has the largest volume and polarizability, twice as large as the positive epinephrine. They exhibit a sudden decrease in diffusivity when the channel heights are $h_1 = 5.7$ nm and $h_2 = 13$ nm, respectively. From $(\text{polarizability})_1/(\text{polarizability})_2 = (h_2/h_1)^6$, we would not expect $h_2/h_1 = 2$ as observed. In view of the uncertainties, though, we can compare the effect of polarizability on the diffusivity itself. Histamine and epinephrine have polarizability differing by 50%, while their diffusivities differ by more than an order of magnitude. On the contrary, cefazolin and 3-aminosalycilic acid have very similar diffusivities, while their respective polarizability differs nearly by a factor of 3. Although these arguments cannot completely rule out non-electrostatic adsorption at the ultra-nanoscale, they do make it not very likely.

Summarizing, it appears that the model in Eq. (2), including electrostatic, HS, and hydrodynamic interactions of solute molecules with the channel walls, is unable to provide even a qualitative description of the diffusivity of charged molecular species in nanochannels. At the ultra-nanoscale, diffusivity suddenly drops in a charge-independent fashion, which cannot be explained by repulsive, HS-like molecule-wall forces or hydrodynamic interactions, and seems hard to justify with attractive, non-electrostatic forces. Other studies have demonstrated transitional behavior at the ultra-nanoscale for solvent transport through carbon nanotubes, but with antipodal consequences: a dramatic increase in flux[34]. As the acceleration was attributed to the hydrophobic nature of the carbon nanotube interiors, this may provide further insight into the marked decrease in transport observed in our system.

A key aspect of the experimental system neglected by the models is the finite volumes of the molecules involved. When the finite size of solute, ions, and solvent molecules becomes a relevant parameter, the mean-field models used here break down, especially at the ultra-nanoscale. It is known from the literature that the volume of ions in solution has a strong effect on the electrostatic potential, when $h$ decreases relative to the Debye length[35,36] and the surface charge increases. If the channel walls are strongly negatively charged, the point-like cationic density $c(z) = c_0 \exp[-q\psi(z)/k_B T]$, where $c_0$ is the bulk concentration, can increase boundlessly, which is not true when each ion occupies a finite volume $a^3$. In this case, $c$ cannot increase beyond[37] $c_{max} = 1/a^3$. The cationic concentration then saturates in the vicinity of the charged surface. However, accounting quantitatively for these effects on the electrolytes and investigating the transport of charged or neutral molecules against the electrostatic background of finite-size ions is non-trivial and will be a topic for future investigations.

We recognize that other factors such as solvent mobility[38], solute solidification[39], or molecular polarization[40] may play a role

in the interpretations of our results. Thus, it is also possible that understanding the universal, charge-independent drop in diffusivity at the ultra-nanoscale will require the use of atomistic models.

Although we are not yet able to fully understand the transport of solutes in our membranes, the present investigation exemplifies the importance of precisely controlling the geometry of the membranes down to the nanometer and below. Various studies[41,42] had previously investigated the effect of electrostatics on nanoscale transport by varying the ionic strength of the solution, and thus the Debye length, in nanochannels of a fixed larger size (i.e., 50 nm). Although seminal and noteworthy, these works could not probe the increasing complexity of charged fluids at an increasing level of physical nano-confinement (<5 nm), where fluctuations in density, viscosity, and diffusivity have been observed[12] to become dominant. All of these phenomena are likely to play a significant role in molecular transport, and cannot be experimentally reproduced through the modification of the number of ions in solutions. In this context and to our knowledge, this experimental study is unprecedented; a detailed analysis of the transport behaviors for diffusing solutes with different charges within a scaled series of nanochannels ranging from the sub-micron to ultra-nanoscale.

The membrane architecture allowed reliable quantification of molecular transport from directly measured mass flux as opposed to computational or microscopic techniques. Our unexpected results are relevant to regulated transport by organic solute carriers across cell and nuclear membranes. Lipid bilayers are impermeable to most essential molecules and ions, and solute carriers are necessary to facilitate organic molecular transport. Findings regarding a charge-independent transport provide insight into the carrier's ability to mediate uptake of a broad range of substrates, from bile acids to steroid conjugates[2]. For analyte sorting[10] and nanofiltration[20], charge-based selectivity at the ultra-nanoscale offers new separation parameters, even for neutral molecules. A key demonstration in our study was the linearization of cation release by ultra-nanoscale confinement. Steady transport can be advantageous for drug delivery[27], and this work allows its extension to cationic drugs. The results are pertinent to a number of other fields, including water desalination[43], tissue engineering[44], small sample manipulation[45], nanogap capacitors[46], and energy conversion[47].

In conclusion, we have presented an unprecedented experimental characterization of solute diffusion in a scaled series of slit-nanochannels. Comparative analysis of molecular diffusion through 2.5–250 nm nanochannels demonstrated different transport regimes depending on channel size, molecular charge, and molecular volume. At the nanoscale, neutral molecules were observed to behave as if they carried an effective charge, though with added complications due to their different affinity for water. Further, an abrupt drop in diffusion for all molecules, irrespective of their charge, was observed at the ultra-nanoscale (channel height <5 nm), a finding that cannot be simply explained in the context of electrostatic, HS, and hydrodynamic interactions. Our results hint at a counterintuitive effective charge scenario at intermediate nanometric length scales, and open a window on a puzzling charge-independent transport that challenges current theory on nanoconfined diffusion. The latter regime was characterized by an abrupt drop in flux and transport linearization. This study is a first step into the unique fluidic environment of the ultra-nanoscale, which may offer profound insight on molecular motion in highly confined systems.

## Methods

**Membrane fabrication**. The membranes were fabricated using standardized industrial processes at a commercial manufacturer as previously described. Each silicon chip had dimensions of 6 by 6 by 0.750 mm (length, width, and thickness) and enclosed large numbers of monodispersed nanochannels (over 50,000 mm$^{-2}$). The fabrication protocol began with a Silicon-On-Insulator (SOI) wafer 200 mm in diameter. This wafer had a device layer of 30 μm, buried oxide layer of approximately 1 μm, and a handle layer of 700 μm. A sacrificial tungsten layer was deposited on the SOI, the depth of which dictated the final nanochannel height. This layer was patterned through photolithography and capped by 20 μm of silicon nitrate. Rectangular outlet microchannels were created vertically with respect to the SOI through a dry etch process through the silicon nitrate layer. The microchannel formation was arrested by reaching the buried oxide layer, exposing the outlets of the nanochannels. The macrochannel inlets, with a dimension of 200 μm by 200 μm, were created by etching the opposite surface of the wafer until reaching the underside of the buried oxide layer. A deep silicon etcher was used to pierce the buried oxide layer and expose the sacrificial layer from beneath, revealing the inlets of the future nanochannels. From this point, a continuous path was created through hydrogen peroxide etching of the sacrificial material to form a contiguous pathway of inlet macrochannels, inlet microchannels, nanochannels, and outlet microchannels. The total number of nanochannels created was consistent across the size range (2.5–250 nm) utilized in all experiments in this manuscript. The capping dielectric stack, and hence the internal stress associated with these dielectrics, was consistent for nanochannel sizes between 2.5 and 40 nm. The stack thickness was far greater than the nanochannel height, leading to a nearly identical structure and geometry for all devices. The thickness was changed for the 250 nm channels, as they required a different integration scheme. The stress was high enough to keep the ceiling of the channels tensile in all cases. Following the sacrificial etching process, IPA was substituted for water prior to drying to minimize surface tension and prevent channel collapse. The resulting slit-channels, presenting a highly defined and precise geometry with sub-nanometer size tolerances, were parallel to the membrane surface and orthogonal to the inlet and outlet microchannels; a design promoting both high channel density and physical robustness (see Fig. 1a). For additional details, refer to Supplementary Notes 1–4.

**Molecule diffusion experimental setup**. Diffusion experiments were performed at controlled RT (23 ± 0.2 °C) with at least three replicates for each channel size: 2.5, 3.6, 5.7, 13, 20, 40, 250 nm. A custom robotic carousel coupled with an Agilent Cary 50 UV/Vis spectrophotometer was used to periodically measure UV absorbance (every 10 min) in the sink reservoir to determine the concentration of histamine ($\lambda_{HA}$ = 230 nm), epinephrine ($\lambda_{EP}$ = 279 nm), aspirin ($\lambda_{AP}$ = 275 nm), L-phenylalanine ($\lambda_{PA}$ = 257 nm), cefazolin ($\lambda_{CF}$ = 280 nm), and 3-aminosalicylic acid ($\lambda_{3aa}$ = 315 nm). Data were normalized with respect to the absorbance at time = 0 and the cumulative release was calculated using absorbance vs. concentration standard curves. Additional details can be found in the Supplementary Notes 5–7. Low and high ionic concentration ($I_L$ and $I_H$, respectively) were obtained as solely pure Millipore water ($I_L$) and Millipore water with 137 NaCl ($I_H$) solution. pH was adjusted (pH 4 for histamine and pH 7 for cefazolin) for each solution for a better comparison with the previously collected data.

**Streaming potential measurements**. The experimental setup for the streaming potential's measurement is represented in Supplementary Fig. 10 and described in Supplementary Note 16. Briefly, a syringe (60 ml) associated with a pressure controller (PC3, Alicat Scienic, AZ) was used to apply a differential pressure across membranes. Streaming potential measurements between two Ag/AgCl electrodes were recorded with an electrochemical analyzer (CHI621D, CH Instruments) at the pressures of 0, 5, 10, and 15 psi.

**Mathematical model**. See Supplementary Notes 8–15.

**Data availability**. All other data supporting the findings of this study, including computer codes, are available from the corresponding author upon reasonable request.

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

## Acknowledgements

We thank Dr. Alma Zecevic and Dr. Lidong Qin for helpful discussions and Randy Goodall and Lee Hudson for their support. We also acknowledge financial support from the Center for the Advancement of Science in Space (CASIS) GA-2013-118, CASIS GA-2014-145, funding from the Houston Methodist Research Institute, NIH R21 GM 111544, NanoMedical Systems, and the Nancy Owens Memorial Foundation. Membranes were provided by NanoMedical Systems.

## Author contributions

A.G. conceived the study. A.G., G.B., E.Z., G.C., and A.P. designed the experiments; G.B., R.L.H., E.Z., Z.S., P.J., and N.D.T. performed the experiments; A.G., M.F., and S.H. co-invented, co-designed, and contributed to fabrication of the nanochannel membranes; G.B., N.D.T., R.L.H., E.Z., M.F., A.P., and A.G. analyzed the data; A.G. and D.D. contributed reagents/materials/analysis tools; G.B., N.D.T., A.P., A.G., R.L.H., C.S.F., G. C., and E.Z. wrote the paper and designed the figures.

## Additional information

**Competing interests:** A.G., S.H., and M.F. disclose a financial interest in NanoMedical Systems, Inc., Austin, TX. The remaining authors declare no competing interests.

