## [Peer Review File · Nature Communications]

Reviewers' comments:

Reviewer #1 (Remarks to the Author):

This paper reports the diffusive transport of Angstrom sized molecules through silica nano-channels with surface charge. Quite surprisingly, they find the transport to be insensitive to the molecule charge when the channel height is between 2 to 4 nm. The experiments are done carefully and the result is novel.

However, I question the mechanism the authors proposed--that electro-neutrality is broken down as the finite volume of the molecule replaces the counterions. The molecules are at most several angstroms in size. It is hard to believe that the counterions cannot enter a 2 nm or larger nanochannel. I believe a more likely answer is the van der Waal force between the molecule and the nanochannel. Below 5 nm, vdW force should dominate over electrostatic forces and I suspect that the dramatic drop in the flux rate at small channel height is due to vdW forces. These vdW forces adsorb the molecules onto the channel wall and significantly decrease their diffusivity.

If vdW adsorption onto the nanochannel is true, the more polarizable molecules should adsorb more and hence exhibit a lower transport rate. Unfortunately, I was unable to determine the diffusivity from the authors' raw data to determine the diffusivity of each molecule. Since the experiments are done at different concentrations for each molecule, I cannot compare the transport rate across all molecules. I urge the authors to see if there is any correlation between polarizability of the molecules and their diffusivity. Their molecules are all roughly the same size and if their diffusivity does vary significantly, the vdW mechanism would be more likely than the counterion blockage mechanism.

Reviewer #2 (Remarks to the Author):

This paper reports on an extensive analysis of drug release by a nanofilter device with a wide range of nanochannel dimensions. The device has already been reported a number of times by the authors, but the present range of nanochannel dimensions, and the smallest dimensions used, make the study of potential interest.

The claim of the paper is, that the data show a novel type of behaviour. I find this statement unsubstantiated by the data and their analysis. If the authors can provide better controls and provide a better model to confirm the claim of 'distinct break in molecular transport', the measurements might provide some new information. I however have seen the main part of the phenomena already observed (and better explained!) in the work of Plecis et al. from 2005 mentioned below.

My main comments:

- The transport data shown in figure 2 for all substances show a decrease in molecular flux with increasing nanochannel height (channels of 5 to 200 nm). This behavior is not expected from straightforward reasoning. Taking for example positive ions, the concentration of the positive ions in the electrical double layer (only a few nm thin at 50 mM salt) is expected to be maximally three times larger at the maximal wall potential of -25 mV (Boltzmann statistics). In the larger channels the bulk diffusion contribution of the positive ions will therefore be much larger than this surface contribution. Furthermore, this enhancing effect would be present for the positive ions only. The flux of the neutral substances and the negative mesalazine however is also decreasing with increasing channel height.

In my opinion, as explained below) the authors provide no proper explanation for this phenomenon of decreasing molecular flux, which is observed for all substances except cefazoline.

In the model as introduced in the supplementary information, if I understand it well, the authors maximize the number of ions in the nanochannel to three times the surface charge number, and in figure S19 thus can show a decreasing 'normalized number of ions' in the channel. Assuming such a maximal number of ions in the entire channel to me makes no sense at all, which makes this explanation quite unconvincing for me.

Summarizing, this part of the data remains unexplained for me. As alternative explanation, for example a limitation of the diffusional flux towards the nanochannels via the connecting membrane microchannels can be imagined. The limiting influence of this part of the diffusion chain will namely increase with increasing nanochannel height.

- Considering the lower diffusion rates in the smallest nanochannel, adsorption should be considered as possible cause.

- The authors missed out on some important references in the field: Plecis et al., Nano Lett. 5 (2005) 1147, and Schoch et al., Phys.Fluids 17(2005)100604; Scoch et al. APL 86(2005)253111. In these references, especially Plecis et al., the influence of the double layer on ionic transport in a nanochannel is experimentally and theoretically investigated; interestingly, Plecis et al. also find lower transport than expected at large double layer extension, for which adsorption is offered as the cause; finally, good reviews on nanofluidic transport are Sparreboom et al, Nat. Nanotech 4 (2009) 713 and Schoch et al., Rev.Mod.Phys. 80 (2008) 839.

- The authors measure the channel resistance with a multimeter. This is a basic experimental error, as electrode polarization consuming will make the measurements invalid. Proper AC measurement in such a case can be found e.g. in the paper of Schoch, Phys.Fluids (2005) 100604.

- In table 1 the authors mention the different experimental parameters. Some analytes are measured at pH = 3, and others at pH 4 or 7. The wall charge density in these cases, and thereby the surface potential, will be quite different as a result: about zero at pH = 3, small at pH=4 and larger at pH = 7. Furthermore details are missing on how the pH was established in these solutions. The wall charge densities will have a large effect on system behaviour and data interpretation.

Other remarks:

- What is logD in Table 1?

- The text should explain that the inverse exponential behavior in e.g. Figure 2a is caused by depletion of the upper reservoir

- Units should be provided for transport rate and transport flux; confusingly, the SI calls this "release rate".

- It is at present very hard to find the width and length of the nanochannels in the text, as well as the width and length of the microchannels that lead to the nanochannel arrays. This is essential for the data interpretation, as it would allow understanding the role of the different sections in the diffusion chain.

- conductance and conductivity are confused throughout

- for the explanation of the behavior observed in figure 12, concentration polarization at the channel entrances should be considered. The explanation as now offered in the SI lines 329 and further is not understandable for me, as the concentration distribution inside the channel is not modified on the application of a current, i.e. ions do not need to be stored.

Reviewer #3 (Remarks to the Author):

April 21, 2017

Stephen Levy

Associate Professor

Physics Department

Binghamton University, SUNY

Review: Unexpected and distinct break in molecular transport at the ultra-nanoscale

The authors measure the diffusion of small drug molecules through nanofluidic slit-like structures with depths ranging from several to hundreds of nanometers. The concentration of a drug molecule is measured by UV absorbance. They observe a rather abrupt change in the transfer rate of the small molecules that diffuse through the nanoslit as the depth is reduced below ~ 5 nm that appears to be independent of the charge of the small molecule. The authors investigate the diffusive rate at lower and higher ionic strength and measure the resistivity of the nanoslits as a function of slit depth. They infer that the physics mechanism explaining these results is the electrostatic penalty for excluding counter-ions from inside the slit due to the difference in volume between the drug molecules and the salt or water counter-ions.

The authors make interesting measurements using well-defined and well-fabricated nanofluidic structures. I believe that the results are important and interesting to the nanofluidic transport community. The results are often not presented in a clear manner. It is not clear that the physics is well described. There are several issues that should be addressed before publication. I am not particularly convinced by the explanation offered for the observed behavior but the results and interpretation are clearly worth of publication once the following issues are addressed.

This manuscript would greatly benefit from a physical and mathematical description of both diffusion through the nanoslits and a model for the charge distribution, and electric potential, in the nanoslits at different ionic strengths and pH values. The authors completely ignore variation in the zeta potential of silica as a function of pH. The surface chemistry is well known and should be discussed since they measure diffusion for the drug molecules at widely varying pH values (1).

The authors present a mathematical model in the methods section that looks like the Poisson-Boltzmann equation (2). However, a second term is added to the model that I have never seen in the Poisson-Boltzmann equation. This second term should be motivated or referenced and explained. They claim that this term is only important in nanochannels that are not electroneutral. However, the term does not explicitly depend on the nanochannel surface charge or dimensions.

The authors attempt to calculate the electrostatic penalty for a larger molecule to enter the nanoslit region. They claim that this explains the observed results. However, it is not clear why the authors only consider the electrostatic energy as opposed to the Helmholtz free energy. The latter contains entropic contributions due to the rearrangement of the counter-ions that play a prominent role in

the proposed explanation (3) . It is not obvious that the authors' explanation is even reasonable when this entropic contribution is ignored. The model also does not take into account that the zeta potential is different for the silica and silicon nitride surfaces of the nanoslit. The authors should address this and explain why it may not matter.

The authors make almost no attempt in the manuscript to define the dependent variables that are measured and plotted in Figure 2. I was rather frustrated attempting to discern the meaning of the 'cumulative mass transport' (units of mass), 'molecule transport rate' (no units given), and molecule flux (no units given). One reason it was difficult to comprehend these variables results from the authors not mentioning how concentration or mass is measured until the 'Methods' section (UV absorbance on line 306). The main experimental method should at least be briefly defined in the main part of the manuscript (one might even contend that it should be stated in the abstract).

I then assumed that the 'transport rate' should be the slope of the mass transport vs time plot. However, the rate is plotted for each nanoslit height. Since it is obvious from Figures 2a, 2d, and 2g that the rate is not constant, it was not clear to me whether they had plotted the average rate or a weighted average rate or something different in Figs 2b, 2e, and 2h. After reading the supplemental information it became clear that they only plotted the rate for the linear portions of the data shown in Figs 2a, 2d, and 2g. This should be stated in the main text or less preferably the reader should be referred to the appropriate section of the supplementary information for a description of the variables plotted.

Unfortunately, the authors do not present any guidance to the reader as to what the expected shape of the signal of mass transport vs time should be (in the bulk regime). I assumed that it should be $A * [1 - \exp(-t/B)]$. The authors state that the 'mean cumulative releases' follow an inverse exponential profile. This should be stated more clearly (and it would help if the authors refer to the variable that they actually plot, which is cumulative mass transport).

There are many interesting features in Figure 2. The authors refer to some of these but not all. I do not understand why the cumulative mass transport in Fig 2a does not increase as the channel height is increased in the range from 20 to 200 nm. The authors seem to imply that this is not yet the 'bulk' regime but it is hard to see how electric double layer effects are important in a ~100 nm deep slit with a ~1.4 nm debye length (50 mM NaCl). This should be explained.

It is difficult to tell in Fig 2e whether the transport rate for neutral molecules scales with channel depth. It appears not to in the log-log plot but it is difficult to see clearly. This should be explained.

The authors do note that there is a sharp reduction in the cumulative mass transport for nanoslits with heights of 2.5 and 3.6 nm. This reduction appears for both positive, neutral, and negatively charged drug molecules.

The authors attempt to validate their model by examining the diffusive transport rate at low and high ionic strength. There is a dramatic difference (2 orders of magnitude) in the rate for histamine that is independent of ionic strength when moving from the 3.6 to 5.7 nm channels. It is not at all clear that the mathematical model that the authors propose is able to explain this difference and its independence on ionic strength (especially considering the perhaps factor of 3 difference in volume between the histamine and sodium or chlorine ions). The authors seem to present this result as validating their model, however. I could not understand how the authors made the plot in Figure S19 that purports to validate their conclusions.

The authors also examine the resistivity of the nanofluidic slits in different ionic conditions. I was extremely puzzled by the data shown in Fig 4a. The authors refer to the data in the caption as 'conductivity'; however the axis label refers to 'conduction' with units of S. As the authors know, the difference between conductivity (with units of S/m) and conductance (with units of S) refers to the geometrical factor of the nanoslit (area / length). It is impossible to interpret the data in this figure with this distinction not being made clear. Since the measured value seems to plateau as a function of slit height I would presume that they are measuring conductivity (S/m). However, I am reluctant to make that conclusion given that the plot axis is labeled conductance (S). The authors should attempt to explain all the differences shown in this Figure between monovalent and divalent solutions and between low and high salt concentration. The features are bizarre.

There is a column labeled 'logD' in Table 1. Presumably this refers to the distribution coefficient but this should be explained.

The mathematical model in the methods states that model assumes the walls are charged with a surface density of -0.06 C / m^2 . Does this correspond to the zeta potential that is listed for silica (-28 mV)?

Equation 3 in the supplementary information is the Debye-Huckel approximation to the PB equation but does not show the extra second term that is shown in the mathematical model section of the main manuscript. Equation 9 of the supplementary information does contain the extra second term but again without a reference or derivation. Why this discrepancy?

1. Behrens, S. H. & Grier, D. G. The charge of glass and silica surfaces. *J. Chem. Phys.* 115, 6716 (2001).
2. Schoch, R. B., Han, J. & Renaud, P. Transport phenomena in nanofluidics. *Reviews of Modern Physics* 80, 839 (2008).
3. Andelman, D. in *Soft Condensed Matter Physics in Molecular and Cell Biology* 97–122 (Taylor & Francis, 2006)

Reply to the reviewer's comments.

We are pleased to submit a revised version of our paper NCOMMS-17-06341 now titled '*Stepping down to the ultra-nanoscale: unexpected behaviors in molecular transport through size-controlled nanochannels*'. Thanks to the reviewers' in-depth reading and pointed criticisms, we have been able to vastly improve the analysis of our results, as well as to greatly clarify the relevance of our findings against the background of current knowledge about nanoscale transport. As a consequence, the article has been extensively rewritten, a detailed model study has been performed, and a much clearer understanding has been achieved. In particular, a modified analysis of the experimental data and their comparison with theoretical descriptions has allowed us to emphasize the novelty of our results, especially concerning neutral solute species and the charge-independent transport at the ultra-nanoscale. Our replies to the reviewers will therefore reflect the extensive changes that will be evident in the present form of our paper. Here below a point-by-point reply to the comments is listed.

Reviewer #1:

Comment 1: This paper reports the diffusive transport of Angstrom sized molecules through silica nano- channels with surface charge. Quite surprisingly, they find the transport to be insensitive to the molecule charge when the channel height is between 2 to 4 nm. The experiments are done carefully and the result is novel.

However, I question the mechanism the authors proposed--that electro-neutrality is broken down as the finite volume of the molecule replaces the counterions. The molecules are at most several angstroms in size. It is hard to believe that the counterions cannot enter a 2 nm or larger nanochannel. I believe a more likely answer is the van der Waal force between the molecule and the nanochannel. Below 5 nm, vdW force should dominate over electrostatic forces and I suspect that the dramatic drop in the flux rate at small channel height is due to vdW forces. These vdW forces adsorb the molecules onto the channel wall and significantly decrease their diffusivity.

Reply 1: We appreciate the comment of the reviewer. We feel that we should first of all clarify a possible source of misunderstanding: by channel electroneutrality, or breakdown of it, we meant to designate the large imbalance between cations and anions that exists inside the nanochannel, when there is a strong overlap of the EDL. Of course, this charge is always compensated by the silica surface charge at equilibrium. Therefore, we do not propose that channel electroneutrality is broken down as the solute molecules enter the channel—electroneutrality within the solution breaks down as soon as the EDLs from both sides of the channel overlap, so that—in our experimental conditions—a large cation excess (“enrichment”) is established. Due to this cation excess, anions and negative molecules are prevented from entering the nanochannel, which is the mechanism (“exclusion”) at the origin of “gated transport” in nanochannels. Our revised analysis of solute diffusivities, which eliminates from the experimental measurements the effects of all membrane components other than the nanochannels, now clearly shows the enrichment of positive solutes, and the exclusion of negative and neutral solutes. The novel and surprising aspects of our experiments are: (1) the exclusion of the neutral phenylalanine (expected to be

insensitive to the electrostatic potential in the solution); (2) the sudden a charge-independent decrease of the diffusivity for neutral, positive and negative molecules at the ultra-nanoscale (which appears as an increase of the exclusion process for negative solutes; and as an abrupt transformation of enrichment into exclusion for positive solutes). We have now clarified, and in part quantified, our proposal that all solute molecules, irrespective of their “bare” charge, assume an “effective charge”, by replacing a large number of positive counterions in solution, proportionally to their volume. We argue that this excess charge is indeed negative. We also show that introducing a particle-wall repulsive (hard sphere) interaction does not provide a quantitative understanding of the ultra-nanoscale behavior of all diffusing species, irrespective of their charge.

Concerning the reviewer’s remark about vdW forces, we believe that our data show that the latter cannot be dominant over electrostatic effects. Indeed, consider aspirin: this is a neutral molecule, which has the same qualitative behavior as the positively charged histamine and epinephrine. The reason for this seemingly surprising observation is that aspirin is the only hydrophobic molecule in our set. Therefore, it has a tendency to segregate from the aqueous solution and remain close to one of the channel surface, similar to a positively charged molecule. Being hydrophobic in nature, the effective attraction to the channel walls has an entropic component (water molecules are released from the cage surrounding each aspirin) and an energetic one—precisely due to vdW attraction to the channel surface. There is no reason why the vdW attraction should increase further when the channel height is around 5 nm, and the ratio of the channel height to the molecular diameter is approximately 7. To further support this picture, we stress that aspirin behaves as the positive histamine: attraction to the channel surface for large channels, then drop in transport at the ultra-nanoscale. Phenylalanine, which has the same polarizability (see below) and is as neutral as aspirin, but not hydrophobic, behaves instead like a negative molecule. The manuscript now includes these considerations regarding the role of vdW forces in the discussion section.

Comment 2: If vdW adsorption onto the nanochannel is true, the more polarizable molecules should adsorb more and hence exhibit a lower transport rate. Unfortunately, I was unable to determine the diffusivity from the authors' raw data to determine the diffusivity of each molecule. Since the experiments are done at different concentrations for each molecule, I cannot compare the transport rate across all molecules. I urge the authors to see if there is any correlation between polarizability of the molecules and their diffusivity. Their molecules are all roughly the same size and if their diffusivity does vary significantly, the vdW mechanism would be more likely than the counterion blockage mechanism.

Reply 2: The suggestion of this reviewer is very interesting. Unfortunately, our data cannot confirm the role of vdW and if anything, they seem to indicate that they are not a major effect. Our conclusion comes from the very argument that the reviewer suggests. The values of polarizability per unit volume (we added these values to Table 1 in the paper) are very close to a constant, equal to 0.12, for all the molecules used in this study. vdW interactions are known to

scale as polarizability divided by the distance to the power 6. The negatively charged cefazolin has the largest volume and polarizability, twice as large as the positive epinephrine. They exhibit a sudden decrease in diffusivity when the channel heights are $h_1 = 5.7$ nm and $h_2 = 13$ nm, respectively. From $(\text{polarizability})_1/(\text{polarizability})_2 = (h_2/h_1)^6$, we would not expect $h_2/h_1 = 2$ as observed.

On the other hand, electrostatic attractive and repulsive interactions with the negative silica wall cannot be neglected, and it is not clear how vdW interactions compare with them in such complex systems.

Concerning correlations between polarizability and diffusivity, we can notice that histamine and epinephrine have polarizability values differing by 50%, while their diffusivities differ by more than an order of magnitude. On the contrary, cefazolin and 3-aminosalicylic acid have very similar diffusivities, while their respective polarizability differs nearly by a factor of 3.

In light of the reviewer's comment, for clarity of presentation, we reworked the Figures, removed the normalized graphs in Figure 2 and plotted the effective diffusivity (D) [cm^2/s] for all molecules in Figure 3. Effective D values were obtained by fitting the cumulative release curves with the extended Fick law of diffusion:

$$n_s(t) = \frac{V_d V_s}{V_d + V_s} \Delta c_0 \left(1 - e^{-\frac{t}{\tau}}\right)$$

where $n_s(t)$ is the drug amount in the sink, V_d and V_s are the drug reservoir and the sink volume respectively, Δc_0 is the initial concentration difference, t is time, and

$$\tau = \frac{V_d V_s}{V_d + V_s} \frac{1}{P}$$

P is the permeability of the system. The diffusivity (D) is obtained as:

$$P = \left(\frac{L_i}{w_i h_i N_i D_{bulk}} + \frac{L_n}{w_n h_n N_n D_n} + \frac{L_o}{w_o h_o N_o D_{bulk}} \right)^{-1}$$

where the indices “ i ”, “ n ” and “ o ” refer to “inlet”, “nanochannels” and “outlet”, respectively, N_i , N_n and N_o are the number of micro- or nanochannels in the three respective regions, and D_{bulk} is the bulk diffusivity. For clarity, the derivation of P and D is now presented in the manuscript and detailed in Supplementary Information.

Reviewer #2

Comment 3: This paper reports on an extensive analysis of drug release by a nanofilter device with a wide range of nanochannel dimensions. The device has already been reported a number of times by the authors, but the present range of nanochannel dimensions, and the smallest

dimensions used, make the study of potential interest.

Reply 3: Nanochannel membranes possessing a similar structure have been previously reported by our group in the context of biomedical applications. However, the achievement of membranes possessing dense arrays of 340,252 nanochannels identical to each other in size and geometry and with 2.5 nm dimensions with a few Å dimensional tolerances is unprecedented. We feel that, within itself, this achievement is significant. Investigations of nanoscale transport have always been limited by unsurmountable technological challenges in generating reliable membranes presenting a defined number of reproducible channels with tightly controlled shape, geometry and sizes. Experiments adopting polymeric membranes, have been limited by intricate pore geometries, poor reproducibility or high channel tortuosity¹. Studies leveraging microfabricated structures have suffered by the difficulty to fabricate nanofluidic systems with a high number of channels. As such, most investigations have relied on experimental data collected with a few channels and minuscule flow outputs². More recent studies using carbon nanotubes, alumina, silicon or titania nanoporous films have similarly suffered from inconsistent channel and pore dimensions, leading to significant uncertainties in the theoretical interpretation of results^{3–5}. It is not by chance that studies investigating electrostatics in nanochannels (as is the case of Plecis *at al.*⁶) have relied on relatively large nanochannels (50 nm, in the case of Plecis *at al.*) and generated overlapping EDLs by modifying the ionic strength of the solution rather than changing the channel size. In this manuscript, we introduce for the first time a technology affording nanochannel membranes identical in architecture to each other, but presenting precise nanochannels from 2.5 to 250 nm that constitute an ideal standard for the study of scaling properties of molecular transport from the ultra-nano (<5 nm) to the sub-micron scale (250 nm). In this work, we had the opportunity to leverage this technology for a systematic and comprehensive study of the scaling properties of diffusive transport of molecules, by changing the channel size, rather than the ionic strength or property of fluids. This allowed us to experimentally account for the complexity of fluid confined within physical spaces at the lower end of the nanoscale, where molecular volume (even for background molecules) becomes comparable—at most within one order of magnitude—to the size of the system. This complexity cannot be captured experimentally in larger nanochannels for which molecular sizes are negligible as compared to the physical system. To the best of our knowledge, such a scaling study has never been reported due to the unavailability of membranes and represent an innovative component of our work.

1. Gruener, S. & Huber, P. Knudsen Diffusion in Silicon Nanochannels. *Phys. Rev. Lett.* **100**, 64502 (2008).
2. Kim, S., Jinschek, J. R., Chen, H., Sholl, D. S. & Marand, E. Scalable Fabrication of Carbon Nanotube/Polymer Nanocomposite Membranes for High Flux Gas Transport. *Nano Lett.* **7**, 2806–2811 (2007).
3. Huang, H. *et al.* Ultrafast viscous water flow through nanostrand-channelled graphene oxide membranes. *Nat. Commun.* **4**, ncomms3979 (2013).

4. Fukutsuka, T., Koyamada, K., Maruyama, S., Miyazaki, K. & Abe, T. Ion Transport in Organic Electrolyte Solution through the Pore Channels of Anodic Nanoporous Alumina Membranes. *Electrochimica Acta* **199**, 380–387 (2016).
5. Zhang, K. *et al.* Water-Free Titania–Bronze Thin Films with Superfast Lithium-Ion Transport. *Adv. Mater.* **26**, 7365–7370 (2014).
6. Plecis, A., Schoch, R. B. & Renaud, P. Ionic Transport Phenomena in Nanofluidics: Experimental and Theoretical Study of the Exclusion-Enrichment Effect on a Chip. *Nano Lett.* **5**, 1147–1155 (2005).

Comment 4: The claim of the paper is, that the data show a novel type of behavior. I find this statement unsubstantiated by the data and their analysis. If the authors can provide better controls and provide a better model to confirm the claim of 'distinct break in molecular transport', the measurements might provide some new information. I however have seen the main part of the phenomena already observed (and better explained!) in the work of Plecis *et al.* from 2005 mentioned below.

Reply 4: We are thankful for the comments of the reviewer. We acknowledge the body of literature that reports on diffusive transport in nanofluidic systems. Steric interactions were well documented in numerous previous papers such as Smith and Deen^{1–3}, while the exclusion-enrichment effect was described in details by Plecis⁴, Taghipoor⁵ and others. It has to be noted, however, that, to the best of our knowledge, none of these works presented results obtained by continuously varying the nanochannel size in a broad range of dimensions, including what we refer to as the “ultra-nanoscale”. To adopt the same example as used by the reviewer, Plecis *et al.*⁴ investigated the interplay of electrostatics and molecular transport in 50 nm nanochannels, and varied the ratio between channel size and Debye length by varying exclusively the latter (through the ionic strength of the solution.) Although that study was seminal and noteworthy, it could not address the numerous effects of nanoconfinement that depend on the size of the channel, but are weakly dependent on the Debye length. In fact, there is a vast body of literature^{6–9} that shows how at the ultra-nanoscale, properties such as density fluctuations, viscosity and diffusivity are affected by the presence of confining boundaries, in a way that cannot be mimicked by a change of ionic strength. These phenomena are especially important below 5 nm^{6,7}, and are likely to play a significant role in molecular transport. Indeed, the strong (factor of 3) decrease in diffusivity for the neutral phenylalanine that we observed below 40 nm, as well as the even stronger and charge-independent decrease observed for all molecules below 5 nm, were never previously reported.

Our study represents the first investigation of diffusive transport where the scaling effect of size confinement on channel electrostatics and molecular transport is directly measured through a set of nanochannels as small as 2.5 nm, rather than modeled and approximated through variation of fluid properties.

In light of the comment, to further analyze the experimental results, we modified Plecis *et al.*⁴'s

theoretical model, which accounts for exclusion-enrichment effects, by including particle-walls interactions following the approach of Smith and Deen^{2,10}. The model also accounts self-consistently for the variation in silica wall surface charge following changes in the pH and ionic strength inside the channel, using a site-binding model as in the Behrens *et al.*¹¹ work. Briefly, our approach to modeling the diffusion of charged molecules inside *slit*-channels of variable heights, containing a NaCl solution at equilibrium between two reservoirs, consists in computing the electrostatic potential $\psi(z)$ inside the channel, due to a 1-1 electrolyte (NaCl), from the Poisson-Boltzmann equation. Following Behrens *et al.*¹¹, the surface charge on the silica channel walls was self-consistently computed using a site-binding model for silica protonation-deprotonation and requiring equilibrium with the charge in the solution. Once $\psi(z)$ was known, we computed the effective diffusivity of the analytes as $D_{\text{eff}} = \beta D_{\text{bulk}}$, where the solute partition coefficient β is representative of an increase or a decrease of the diffusivity in the nanochannels with respect to the bulk, because of an “electro-steric” effect. The partition coefficient β was defined as the average of the analyte concentration over the channel height, assuming that the diffusing molecules with electric charge q follow a Boltzmann distribution, $c(z) = \exp(-[E + q\psi]/k_B T)$, where E is a hard-sphere potential, and $\psi(z)$ the electrostatic potential of the solution. Following Smith and Deen^{2,10}, neutral molecules should only be sensitive to the steric part of the potential, so that $\beta_{\text{st}} = (h - 2r_s)/h$, which is the average over the channel height, of a hard-sphere potential preventing the center of mass of each molecule to approach the channel walls by less than its molecular diameter, r_s . The model is introduced in the manuscript and fully detailed in the Supplementary Information Section 9.

Although the model qualitatively describes the transport of charged species, three aspects remain unexplained and novel: 1) The diffusivity of neutral molecules depends on their affinity for water (hydrophobicity or hydrophilicity), but overall they behave as if they carried an effective charge: hydrophobic behaves as positive, hydrophilic, as negative; 2) An abrupt drop in diffusivity is visible for all molecules, independently of their charge, at channel heights of just a few nanometers (see the article for details); 3) A constant (independent of channel height) diffusivity is observed for negative charges in the range from 5 and 40 nm; These novel findings have been highlighted in the manuscript.

While at this time we are unable to fully explain 3) we believe it to be a manifestation of exclusion effects. Instead we propose a qualitative model addressing 1) and 2) based on the finite volumes of the ions and solutes, which are usually disregarded: consider a molecule of volume $A^3 = n a^3$, $n > 1$, diffusing in a nanochannel where the cation concentration is near saturation; on entering the channel, the larger molecule replaces n smaller cations; if the molecule carries a charge $q = Ze$, with $Z < n$, on entering the channel the total charge Q will change by $\Delta Q = (Z - n)e < 0$. Therefore, a neutral molecule ($Z = 0$) appears to carry a net effective negative charge $\Delta Q \approx -(A^3/a^3)e$, which would qualitatively explain the behavior of phenylalanine. How this might affect the ultra-nanoscale is not clear. Our hypothesis is now presented in the manuscript with additional information provided in Supplementary Information Section 13.

1. Smith, F. G. & Deen, W. M. Electrostatic double-layer interactions for spherical colloids in cylindrical pores. *J. Colloid Interface Sci.* **78**, 444–465 (1980).
2. Smith, F. G. & Deen, W. M. Electrostatic effects on the partitioning of spherical colloids between dilute bulk solution and cylindrical pores. *J. Colloid Interface Sci.* **91**, 571–590 (1983).
3. Deen, W. M. Hindered transport of large molecules in liquid-filled pores. *AIChE J.* **33**, 1409–1425 (1987).
4. Plecis, A., Schoch, R. B. & Renaud, P. Ionic Transport Phenomena in Nanofluidics: Experimental and Theoretical Study of the Exclusion-Enrichment Effect on a Chip. *Nano Lett.* **5**, 1147–1155 (2005).
5. Taghipoor, M., Bertsch, A. & Renaud, P. An improved model for predicting electrical conductance in nanochannels. *Phys. Chem. Chem. Phys.* **17**, 4160–4167 (2015).
6. Huang, H. *et al.* Ultrafast viscous water flow through nanostrand-channelled graphene oxide membranes. *Nat. Commun.* **4**, ncomms3979 (2013).
7. Ma, M. *et al.* Water transport inside carbon nanotubes mediated by phonon-induced oscillating friction. *Nat. Nanotechnol.* **10**, 692–695 (2015).
8. Qin, X., Yuan, Q., Zhao, Y., Xie, S. & Liu, Z. Measurement of the Rate of Water Translocation through Carbon Nanotubes. *Nano Lett.* **11**, 2173–2177 (2011).
9. Qiu, Y., Ma, J. & Chen, Y. Ionic Behavior in Highly Concentrated Aqueous Solutions Nanoconfined between Discretely Charged Silicon Surfaces. *Langmuir* **32**, 4806–4814 (2016).
10. Deen, W. M. Hindered transport of large molecules in liquid-filled pores. *AIChE J.* **33**, 1409–1425 (1987).
11. Behrens, S. H. & Grier, D. G. The charge of glass and silica surfaces. *J. Chem. Phys.* **115**, 6716–6721 (2001).

Comment 5: The transport data shown in Figure 2 for all substances show a decrease in molecular flux with increasing nanochannel height (channels of 5 to 200 nm). This behavior is not expected from straightforward reasoning. Taking for example positive ions, the concentration of the positive ions in the electrical double layer (only a few nm thin at 50 mM salt) is expected to be maximally three times larger at the maximal wall potential of -25 mV (Boltzmann statistics). In the larger channels the bulk diffusion contribution of the positive ions will therefore be much larger than this surface contribution. Furthermore, this enhancing effect would be present for the positive ions only. The flux of the neutral substances and the negative mesalazine however is also decreasing with increasing channel height. In my opinion, as explained below) the authors provide no proper explanation for this phenomenon of decreasing molecular flux, which is observed for all substances except cefazoline.

Reply 5: In light of this comment, we put extra effort in improving the clarity of presentation. As such, experimental results previously in Figure 2 are now presented in two separate Figures: Figure 2 plots representative cumulative release profiles for a positive, a negative, and a neutral

molecule. Figure 3 now shows the effective diffusivity obtained from the experimental data as detailed above (*Reply 2 to Reviewer 1*) for all molecules tested in comparison with the electro-steric model previously described.

The effective diffusivity of the positive solutes (histamine and epinephrine in Figure 3) presents the following behavior: In large channels ($h > 40$ nm) their diffusivity is approximately a constant, expected to be close to its bulk value. For smaller channels, the effective diffusivity is seen to increase. Finally, for ultra-nanoscale channels the diffusivity drops abruptly by more than an order of magnitude.

Positive molecules tend to diffuse closer to the channel surface. Within the range from 5.7 to 40 nm, this *near-surface* diffusion dominates over the transport of positive molecule in the bulk. This behavior is evident by observing how in the cumulative release curves (Figure 2) completely overlap for channels from 5.7 to 40 nm. Despite an increase in channel size by a factor of approximately 7, the release profile remains tightly consistent, implying that the diffusive transport of the positive species is not dependent on the amount of bulk in the channel, but rather on the surface area. The width of all channels used is 3 μm . Thus, the surface area between 5.7 and 40 nm channels remains nearly identical (1.1% difference). In the calculation of the diffusivity, the release rate is divided by the whole channel cross section. As such, because the release rate remains constant while the channel cross section increases, the effective diffusivity is seen to decrease approximately as $1/h$ from 5.7 to 40 nm.

In 250 nm channels, positive charges can diffuse in the bulk unaffected by the walls. As such an increase in release rate is observed with respect to the channels from 5.7 to 40 nm (Figure 2). The drop in diffusivity below 5.7 nm is presently unexplained. In this regime of confinement, discrete molecular effects become non-negligible and mean-field, continuum type of models such as Poisson-Boltzmann break down, and a more microscopic description becomes necessary, which is beyond the scope of this article. This explanation is now included in the manuscript. For clarity of presentation, mesalazine was removed from the manuscript so that the paper now contains data for 2 molecules for each charge (positive, negative and neutral).

Comment 6: In the model as introduced in the Supporting Information, if I understand it well, the authors maximize the number of ions in the nanochannel to three times the surface charge number, and in Figure S19 thus can show a decreasing 'normalized number of ions' in the channel. Assuming such a maximal number of ions in the entire channel to me makes no sense at all, which makes this explanation quite unconvincing for me.

Reply 6: Thanks to the precious comments received by the three reviewers, the theoretical approach has been entirely reworked and widely improved. The model accounts for exclusion-enrichment and steric effects as well as the modification in silica wall surface charge based on the pH and concentration of ions in the channel. Details are provided above (*Reply 4 to Reviewer 2*).

Comment 7: Summarizing, this part of the data remains unexplained for me. As alternative explanation, for example a limitation of the diffusional flux towards the nanochannels via the connecting membrane microchannels can be imagined. The limiting influence of this part of the diffusion chain will namely increase with increasing nanochannel height.

Reply 7: In order to focus our analysis on the transport across the nanochannels and to isolate and the effect of the connecting microchannel in the diffusion chain, we have extracted the diffusivity of molecules in the nanochannels from the experimental data as presented above (*Reply 2 to Reviewer 1*). By doing so we eliminated the potentially confounding effect of microchannels.

Comment 8: Considering the lower diffusion rates in the smallest nanochannel, adsorption should be considered as possible cause.

Reply 8: We considered adsorption due to WdW interactions as a possible mechanism for the decrease of diffusivity at the ultra-nanoscale in Reply 2 to Reviewer 1, where we explained why we consider this explanation as unlikely.

Comment 9: The authors missed out on some important references in the field: Plecis et al., Nano Lett. 5 (2005) 1147, and Schoch et al., Phys.Fluids 17(2005)100604; Scoch et al. APL 86(2005)253111. In these references, especially Plecis et al., the influence of the double layer on ionic transport in a nanochannel is experimentally and theoretically investigated; interestingly, Plecis et al. also find lower transport than expected at large double layer extension, for which adsorption is offered as the cause; finally, good reviews on nanofluidic transport are Sparreboom et al, Nat. Nanotech 4 (2009) 713 and Schoch et al., Rev.Mod.Phys. 80 (2008) 839.

Reply 9: We are very thankful for the suggestion. Several of these relevant references are now included in the manuscript. We have extensively used Plecis *et al.* to build the model used in the present version of our paper. Concerning their suggestion that adsorption may be relevant to interpret their own results, we wish to point out that in their measurements, 50 nm height nanochannels were used, very far from the ultra-nanoscale ($h < 5$ nm) investigated here.

Comment 10: The authors measure the channel resistance with a multimeter. This is a basic experimental error, as electrode polarization consuming will make the measurements invalid. Proper AC measurement in such a case can be found e.g. in the paper of Schoch, Phys.Fluids (2005) 100604.

Reply 10: We thank the reviewer for this comment. The manuscript was broadly reworked and now includes a new theoretical approach. Based on this, the channels resistance measurements are no longer part of the manuscript.

Comment 11. In table 1 the authors mention the different experimental parameters. Some analytes are measured at pH = 3, and others at pH 4 or 7. The wall charge density in these cases,

and thereby the surface potential, will be quite different as a result: about zero at pH = 3, small at pH=4 and larger at pH = 7. Furthermore details are missing on how the pH was established in these solutions. The wall charge densities will have a large effect on system behavior and data interpretation.

Reply 11: The new theoretical approach accounts for the variation in wall charge density based on different pH and ionic concentration of the solution in the nanochannel full details of the model derivation are provided in Supplementary Information Section 10. The pH of the tested solution was modified by adding negligible quantities of HCl or KOH until the wanted pH was read. No buffer solution were used, which would complicate the system model. The pH of the solution was measured before each the experiment using a Mettler Toledo pH meter. These details are now included in the Supplementary Information Section 6.

Comment 12: Other remarks: What is logD in Table 1?

Reply 12: Similarly to logP, logD provide a measure for the affinity of the molecule with water, or a measure of hydrophobicity. However, logP only accounts for the un-ionized portion of molecules. LogD is inclusive of both un-ionized and ionized species, and therefore is most appropriate for this study. The definition of logD is now included in the caption of Table 1 in the manuscript.

Comment 13: The text should explain that the inverse exponential behavior in e.g. Figure 2a is caused by depletion of the upper reservoir.

Reply 13: The clarification is now added in the manuscript.

Comment 14: Units should be provided for transport rate and transport flux; confusingly, the Supporting Information calls this "release rate".

Reply 14: Appropriate units of measurements are now included in the modified Figure 2 and 3, where cumulative mass release (mg or μg) and effective diffusivity (cm^2/s) are presented.

Comment 15: It is at present very hard to find the width and length of the nanochannels in the text, as well as the width and length of the microchannels that lead to the nanochannel arrays. This is essential for the data interpretation, as it would allow understanding the role of the different sections in the diffusion chain.

Reply 15: The width and length of all channels are now included in the caption of Figure 1, which is descriptive of the membrane structure. However, as described above (*Reply 8 to Reviewer 2*) the manuscript is focused on the effective diffusivity in nanochannels. The contribution of the microchannel in the diffusion chain was isolated during the calculation of the effective diffusivity.

Comment 16: conductance and conductivity are confused throughout - for the explanation of the behavior observed in Figure 12, concentration polarization at the channel entrances should be considered. The explanation as now offered in the Supporting Information lines 329 and further is not understandable for me, as the concentration distribution inside the channel is not modified on the application of a current, i.e. ions do not need to be stored.

Reply 16: We thank the reviewer for this comment. The manuscript was broadly reworked and now includes a new theoretical approach. Based on this, the channel conductance considerations are no longer part of the manuscript.

Reviewer #3

Comment 17: The authors measure the diffusion of small drug molecules through nanofluidic slit-like structures with depths ranging from several to hundreds of nanometers. The concentration of a drug molecule is measured by UV absorbance. They observe a rather abrupt change in the transfer rate of the small molecules that diffuse through the nanoslit as the depth is reduced below ~5 nm that appears to be independent of the charge of the small molecule. The authors investigate the diffusive rate at lower and higher ionic strength and measure the resistivity of the nanoslits as a function of slit depth. They infer that the physics mechanism explaining these results is the electrostatic penalty for excluding counter-ions from inside the slit due to the difference in volume between the drug molecules and the salt or water counter-ions.

The authors make interesting measurements using well-defined and well-fabricated nanofluidic structures. I believe that the results are important and interesting to the nanofluidic transport community. The results are often not presented in a clear manner. It is not clear that the physics is well described. There are several issues that should be addressed before publication. I am not particularly convinced by the explanation offered for the observed behavior but the results and interpretation are clearly worth of publication once the following issues are addressed.

This manuscript would greatly benefit from a physical and mathematical description of both diffusion through the nanoslits and a model for the charge distribution, and electric potential, in the nanoslits at different ionic strengths and pH values. The authors completely ignore variation in the zeta potential of silica as a function of pH. The surface chemistry is well known and should be discussed since they measure diffusion for the drug molecules at widely varying pH values (1). (1). Behrens, S. H. & Grier, D. G. *The charge of glass and silica surfaces. J. Chem. Phys.* 115, 6716 (2001).

Reply 17: We are grateful for the reviewer's comments. Based on these, we have largely improved the manuscript for clarity of presentation as well as in terms of the theoretical interpretation of the experimental data. The model now accounts for exclusion-enrichment and steric effects as well as for the modification in silica wall surface charge based on the pH and concentration of ions in the channel in line with the work from Behrens *et al.* A summary of the model is reported above (*see Reply 4 to Reviewer 2*), in the manuscript and extensively detailed

in the Supplementary Information Section 11.

Comment 18: The authors present a mathematical model in the methods section that looks like the Poisson-Boltzmann equation (2). However, a second term is added to the model that I have never seen in the Poisson-Boltzmann equation. This second term should be motivated or referenced and explained. They claim that this term is only important in nanochannels that are not electroneutral. However, the term does not explicitly depend on the nanochannel surface charge or dimensions. (2). *Schoch, R. B., Han, J. & Renaud, P. Transport phenomena in nanofluidics. Reviews of Modern Physics 80, 839 (2008).*

Reply 18: The reviewer is right. We follow in the present version of the paper the approach of Plecis et al (2005).

Comment 19: The authors attempt to calculate the electrostatic penalty for a larger molecule to enter the nanoslit region. They claim that this explains the observed results. However, it is not clear why the authors only consider the electrostatic energy as opposed to the Helmholtz free energy. The latter contains entropic contributions due to the rearrangement of the counter-ions that play a prominent role in the proposed explanation (3). It is not obvious that the authors' explanation is even reasonable when this entropic contribution is ignored. The model also does not take into account that the zeta potential is different for the silica and silicon nitride surfaces of the nanoslit. The authors should address this and explain why it may not matter. (3). *Andelman, D. in Soft Condensed Matter Physics in Molecular and Cell Biology 97–122 (Taylor & Francis, 2006).*

Reply 19: The reviewer is correct in stating that the entropy of the ions needs to be accounted for. In principle, we would expect that entropic effect favor the ejection of ions from the nanochannel to the bulk solution, which should increase the effective negative charge on the larger molecule, and thus its energetic penalty. However, it is exceedingly difficult to account for all these effects self-consistently. We limit ourselves in this revised version, to investigate a model, based on Plecis et al. and Behrens et al., that takes into account electro-steric effects in the channels as well as the variation of the silica surface charge density based on variation of the ionic strength and pH of the solution. We compute accordingly a solute partition coefficient that through the Boltzmann distribution of the ion-water solution takes entropic effect into consideration.

Both surfaces of the channels are oxidized and exhibit a silica layer at the fluid-solid boundary. Differences exist but are not large. Thus, with the primary objective of gaining a qualitative understanding of the intriguing phenomena observed in the channels, we opted for a simplified model accounting for symmetric surface properties. The main feature of this model is that the electrostatic potential between two parallel walls does not vanish when the distance between the walls becomes small enough. This leads (with our membranes) to the exclusion of anions from the nanochannel. Taking into account the walls asymmetry would only change the position of the

maximum of the potential as well as double the parameters of the site-banding models, introducing quantitative but no qualitative differences. In particular, at the ultra-nanoscale, even the most advanced models fail to account for the complexity of fluid confined within physical spaces whereby molecular volume (even for background molecules) becomes comparable to the size of the system. As such, seeking precise quantitative understanding of the system may not be as relevant considering that gross errors may be done even by considering fluids as a continuum.

Comment 20: The authors make almost no attempt in the manuscript to define the dependent variables that are measured and plotted in Figure 2. I was rather frustrated attempting to discern the meaning of the ‘cumulative mass transport’ (units of mass), ‘molecule transport rate’ (no units given), and molecule flux (no units given). One reason it was difficult to comprehend these variables results from the authors not mentioning how concentration or mass is measured until the ‘Methods’ section (UV absorbance on line 306). The main experimental method should at least be briefly defined in the main part of the manuscript (one might even contend that it should be stated in the abstract).

Reply 20: The section describing Figure 2 was implemented including a brief statement of how the measurements were taken and the reference to the Supplementary Information Section 6-7 for more details.

Comment 21: I then assumed that the ‘transport rate’ should be the slope of the mass transport vs time plot. However, the rate is plotted for each nanoslit height. Since it is obvious from Figures 2a, 2d, and 2g that the rate is not constant, it was not clear to me whether they had plotted the average rate or a weighted average rate or something different in Figs 2b, 2e, and 2h. After reading the supplemental information it became clear that they only plotted the rate for the linear portions of the data shown in Figs 2a, 2d, and 2g. This should be stated in the main text or less preferably the reader should be referred to the appropriate section of the Supporting Information for a description of the variables plotted.

Reply 21: In light of this comment, we put extra effort in improving the clarity of presentation. As such, experimental results previously in Figure 2 are now presented in two separate Figures: Figure 2 plots representative cumulative release profiles for a positive, a negative, and a neutral molecule. Figure 3 now shows the effective diffusivity obtained from the experimental data as detailed above (*Reply 2 to Reviewer 1*) for all molecule tested in comparison with the electro-steric model previously described. We also paid particular attention to provide clear indications of both the variable plotted as well as the method and procedure used throughout the manuscript.

Comment 22: Unfortunately, the authors do not present any guidance to the reader as to what the expected shape of the signal of mass transport vs time should be (in the bulk regime). I assumed that it should be $A*[1 - \exp(-t/B)]$. The authors state that the ‘mean cumulative releases’ follow an inverse exponential profile. This should be stated more clearly (and it would help if the authors refer to the variable that they actually plot, which is cumulative mass transport). There

are many interesting features in Figure 2. The authors refer to some of these but not all.

Reply 22: We appreciate the reviewer comment and the need for extra clarity in the presentation. The shape of fitting function has been explicitly given, the terminology has been made more precise and homogenous, and the discussion of the data has been clarified.

Comment 23: I do not understand why the cumulative mass transport in Fig 2a does not increase as the channel height is increased in the range from 20 to 200 nm. The authors seem to imply that this is not yet the ‘bulk’ regime but it is hard to see how electric double layer effects are important in a ~100 nm deep slit with a ~1.4 nm debye length (50 mM NaCl). This should be explained.

Reply 23: Positive molecules tend to diffuse closer to the channel surface. Within the range from 5.7 to 40 nm, this *near-surface* diffusion dominates over the transport of positive molecule in the bulk. This behavior is evident by observing how in the cumulative release curves (Figure 2) completely overlap for channels from 5.7 to 40 nm. Despite an increase in channel size by a factor of 8, the release profile remains tightly consistent, implying that the diffusive transport of the positive species is not dependent on the amount of bulk in the channel, but rather on the surface area. The width of all channels used is 3 μm . Thus, the surface area between 5.7 and 40 nm channels remains nearly identical (1.1% difference). In the calculation of the diffusivity, the release rate is divided by the whole channel cross section. As such, because the release rate remains constant while the channel cross section increases, the effective diffusivity is seen to decrease approximately as $1/h$ from 5.7 to 40 nm. This is a direct consequence of the enrichment effect: the concentration of positive species increases next to the negative walls, as soon as the electrostatic potential becomes appreciably more negative, and anions are ejected from the channel. The corresponding diffusivity is actually very well described by the enrichment model for channels in the 5.7 - 40 nm range.

The reviewer is correct in pointing out that in a 100 nm channel, such effect should be negligible. As a matter of fact, we do not have measurements between $h = 40$ nm and 250 nm, so that our data do not contradict the reviewer’s expectation.

Membranes with $h = 250$ nm channels do indeed show that positive solutes diffuse in the bulk, apparently unaffected by the presence of the walls. The corresponding increase in cumulative mass release is observed with respect to the channels from 5.7 to 40 nm (Figure 2).

Comment 24: It is difficult to tell in Fig 2e whether the transport rate for neutral molecules scales with channel depth. It appears not to in the log-log plot but it is difficult to see clearly. This should be explained.

Reply 24: The additional work performed, gave us the opportunity to highlight new important findings, previously neglected. The peculiar and previously unreported behavior of neutral molecules is one of these intriguing findings. To improve the clarity of presentation and to better highlight these new observation we have reworked Figure 2 and split the results in two separate

Figures as previously described. Transmembrane transport of the neutral analytes aspirin (Figure 2c) and phenylalanine (see Supplementary Information Section 7) exhibited unexpected features. Both molecules maintained far more linear profiles than the cations (Figure 2a) at channel heights of 5.7 nm and above (first region); however, aspirin appears to behave similarly to histamine, in that its cumulative release is only weakly dependent on channel size. On the other hand, phenylalanine shares behavior with anions. These characteristics are more clearly established when an effective diffusivity is extracted from the release data. The similarity to charged molecules extends to mass transport through 2.5 and 3.6 nm channels, where a clear drop in transport rates is observed. As the concentration of neutral analytes, and thus their release rate, should be unaffected by the charges along the channel walls and in the solution, the similarities that the release of aspirin and phenylalanine share with cations and anions, respectively, are extremely surprising and intriguing. The diffusivity data now presented in Figure 3 provides additional information. Strikingly, the diffusivity of the neutral molecule aspirin is h -dependent and its profile is indistinguishable from that of positively charged species, while that of phenylalanine closely matches the diffusivity of negatively charged molecules. The fact that different neutral molecules can exhibit different diffusion behavior from one another is surprising in itself, but becomes less so when we consider that aspirin is a strongly hydrophobic molecule—this is what the positive value of $\log D$ in Table 1 implies. As such, it may be expected to diffuse close to the channel walls, and thus to exhibit a near-surface diffusion behavior mimicking a positive charge. We will propose an explanation of the completely unexpected behavior of phenylalanine based on an “equivalent charge” model, originating from the finite volume of the molecule, as well as of all the ions in solutions. In short, introducing a large neutral molecule in a channel removes a number of cations and anions proportionally to its volume, thus lowering the total number of charges. In conditions in which the solution is enriched in cations and depleted in anions (the case of the ultra-nanoscale), more cations than anions will be removed, corresponding to adding an equivalent negative charge to the solution. The new findings and a thorough discussion of the results related to the behavior of neutral molecules are now included in the manuscript.

Comment 26: The authors do note that there is a sharp reduction in the cumulative mass transport for nanoslits with heights of 2.5 and 3.6 nm. This reduction appears for both positive, neutral, and negatively charged drug molecules.

Reply 26: We agree with the reviewer. A sharp reduction in diffusivity is observed for all molecules for channels smaller than 5.7 nm, which channel size is still too large to simply dismiss the finding in the context of pure steric effects. We show that these findings cannot be explained in the framework of simple electrostatic models, and we propose an alternative simple, semi-quantitative argument to formalize our (limited) understanding of what happens at the ultra-nanoscale. This is summarized in *Reply 4 to Reviewer 2*, as well as in the manuscript. We stress that a quantitative description the observed behavior requires a theoretical work well beyond the scope of this work.

Comment 27: The authors attempt to validate their model by examining the diffusive transport rate at low and high ionic strength. There is a dramatic difference (2 orders of magnitude) in the rate for histamine that is independent of ionic strength when moving from the 3.6 to 5.7 nm channels. It is not at all clear that the mathematical model that the authors propose is able to explain this difference and its independence on ionic strength (especially considering the perhaps factor of 3 difference in volume between the histamine and sodium or chlorine ions). The authors seem to present this result as validating their model, however. I could not understand how the authors made the plot in Figure S19 that purports to validate their conclusions.

Reply 27: We agree that the results of Figure S19 did not add much to the understanding, and in fact were only confusing. They have been removed. We reiterate that we use now a model based on the Poisson-Boltzmann equation, and compute the effective diffusivity by means of a solute partition coefficient. We show that this model is unable to capture the strong, charge-independent decrease in diffusivity in ultra-nanoscale ($h < 5.7$ nm) channels.

Comment 28: The authors also examine the resistivity of the nanofluidic slits in different ionic conditions. I was extremely puzzled by the data shown in Fig 4a. The authors refer to the data in the caption as ‘conductivity’; however the axis label refers to ‘conduction’ with units of S. As the authors know, the difference between conductivity (with units of S/m) and conductance (with units of S) refers to the geometrical factor of the nanoslit (area / length). It is impossible to interpret the data in this Figure with this distinction not being made clear. Since the measured value seems to plateau as a function of slit height I would presume that they are measuring conductivity (S/m). However, I am reluctant to make that conclusion given that the plot axis is labeled conductance (S). The authors should attempt to explain all the differences shown in this Figure between monovalent and divalent solutions and between low and high salt concentration. The features are bizarre.

Reply 28: We thank the reviewer for this comment. The manuscript was broadly reworked and now includes a new theoretical approach. Based on this, the channel conductance considerations are no longer part of the manuscript.

Comment 29: There is a column labeled ‘logD’ in Table 1. Presumably this refers to the distribution coefficient but this should be explained.

Reply 29: Yes, logD refers to the distribution coefficient. Similarly to logP, logD provide a measure for the affinity of the molecule with water, or a measure of hydrophobicity. However, logP only accounts for the un-ionized portion of molecules. LogD is inclusive of both un-ionized and ionized species, and therefore is most appropriate for this study. The definition of logD is now included in the caption of Table 1 in the manuscript.

Comment 30: The mathematical model in the methods states that model assumes the walls are

charged with a surface density of -0.06 C / m^2 . Does this correspond to the zeta potential that is listed for silica (-28 mV)?

Reply 30: In the previous version of the manuscript, the listed surface charge density was related to silica surfaces in contact with 50 mM NaCl . Our new model accounts for the variation of surface charges at different pH and ionic strength of solutions, rendering the previous parameters obsolete.

Comment 31: Equation 3 in the Supporting Information is the Debye-Huckel approximation to the PB equation but does not show the extra second term that is shown in the mathematical model section of the main manuscript. Equation 9 of the Supporting Information does contain the extra second term but again without a reference or derivation. Why this discrepancy?

Reply 31: The Supplementary Information Section 11 now includes details of the new model derivation. The discrepancy has been removed.

Reviewers' comments:

Reviewer #1 (Remarks to the Author):

The authors have convinced me that adsorption is not an issue for the diffusion of neutral molecules through ultra-fine nanopores, based on the polarizability values now listed in the table. I suspect then the effective net charge they see with these neutral molecules in ultra-fine nanopores is because of the pH in these pores, which is much lower than in the bulk. This would explain why aspirin seems to have a positive charge. Proteins like phenylalanine has two pKa's and it could well be that they become negative at low pH due to dissociation of certain functional groups.

I would like the authors to comment on this possibility. I believe that they have found a new phenomenon but, because they do try to develop a model to explain it, all possible mechanisms should be at least discussed in the published paper.

I can recommend publication if the authors can describe this pH effect in the manuscript.

Reviewer #2 (Remarks to the Author):

General issue:

In general the paper uses many hyperbolic words: "unprecedented", "first of its kind", "unique". As I will show below, experiments like this have been extensively performed in the 1980s and have been reported in a number of papers. In these experiments generally track-etched membranes were used, and the influence of diffusing molecular size and pore size were experimentally studied and theoretically analyzed by a sophisticated model including hindered diffusion on top of steric hindrance and electrical charge (the authors only consider the last two). The only aspect in which the present paper is truly new is in the use of a new geometry (slit-type pores) and probably a better controlled pore homogeneity. That however is all, and theoretically it still lags the 1980 papers. For example, it does not reach the quality and thoroughness of the paper of Deen and Smith "Hindered diffusion of synthetic polyelectrolytes in charged microporous membranes", *J. Membrane Sci.* 12 (1912) 217-237. With respect to the previous paper the paper has much improved, and some fundamental issues have been clarified with respect to the previous version, especially by employing the theory from the Plecis paper. However, when reading the revised version, the reviewer realized that up till now a centrally important aspect of the transport has been neglected: hydrodynamic interactions have been neglected, while static interactions with the wall were accounted for

(electrical forces, Born repulsion and van der Waals forces). The reviewer apologizes he did not notice this before. A central concept in hydrodynamics is that of hindered diffusion by hydrodynamic interaction of the diffusing particle with the wall. In the 1987 review paper of Deen (to which the authors refer), they will find both experimental and theoretical information on the hindered diffusion coefficient, in this paper denoted by K_d . Most interestingly for the present paper, figures 6 and 7 in the Deen paper show experimental data on hindered diffusion of small solutes in tracketched pore membranes with pore radii of 4.5 – 30 nm, in the same range as the present paper. The present paper now needs an added analysis of its data in light of these historical papers that investigate systems of the same size (pore diameter), though different geometry (cylindrical instead of slit-type). The historical papers concerned are mentioned in the Deen paper on pages 1418 and 1419. Especially the paper of Deen and Smith from 1982 performs a thorough analysis of a nanopore system, in which electrostatic effects, steric effects and hindered diffusion are all considered, and the surface charge is even independently measured using streaming current measurements. Notably in these papers, a strongly decreased diffusional transport is observed already when the particle radius is 0.2 times the pore radius. In light of these works I conclude that it is imperative that the authors include hindered diffusion in their model. The values of D/D_∞ for slit-type pores can e.g. be extracted from eqn 32 in the Deen paper.

Separate issues:

I do not understand why the charge of aspirin has such a large influence on the predicted transport rate. The experiments with aspirin have been performed at pH = 3, where the silica is practically uncharged, so that electrostatic effects should be negligible. In the context of the above remark, can the authors in the supplementary information section 10 provide a table with the model-predicted surface potential at the different pH values used? The model presented in section 10 of the supplementary information seems to assume that the diffuse layer potential ψ_d is located at $z=h$. This implies that h denotes half the channel height, while it denotes the full channel height in the rest of the paper. This should be clarified. It also possibly has led to a mistake with a factor of two. The effective charge argument of the authors (supplementary information 13) is impossible to follow. In my opinion the underlying physics is erroneous. As it is now, it severely undermines the believability of the entire paper, as it (at least for me) shows a rather casual attitude to theory development. In the generally used model, the z -dependent anion and cation concentration in the channel is purely determined by the Boltzmann distribution between bulk and channel, based on differences in the local (electrical, van der Waals,...) potential. Concentrations of other species, when equal to their bulk concentration, do not figure as they do not cause changes in the interaction potential. An argument on the presence of other molecules can only be constructed when they influence the local energy state (potential) of the molecule under consideration at the walls to a different extent than the local energy state in the bulk. That can be the case for example when they adsorb to large extent and change the local environment at the wall. In that case however other equations should be used than the ones now used and a great number of assumptions and additional measurements will be needed to prove the hypothesis.

Small points:

Page 2 lines 42/43: "Importantly, experiments distinctly showed that the diffusive transport of cationic, anionic, as well as neutral species at the ultra-nanoscale is dominated by electro-steric

effects and charge density.” “electro-steric effects” is an unknown term that needs definition, for example as “electrical and steric”. The “...and charge density” raises the question: charge density of what: of the diffusing molecule or the pore wall? If so, how is this different from the “electro-” part of the “electro-steric” effect? If it just adds a further qualification, the sentence should reflect this.

Page 3 lines 66/67: “In this regime, all molecules appear to possess an electric charge, either actual or effective, so that transport of all types of solute is determined by the interplay of electrostatic, steric and volume-related effects.”: what is the difference between steric and volume-related effects?

Table 1: are the bulk diffusion coefficients comparable to the literature-reported values for bulk diffusion?

Page 7: At 50 mM NaCl the Debye length is about 2 nm. I am surprised that the EEM produce the predicted dominance of surface diffusion even at 40 nm, when $h/L_D = 20$... Can the authors provide a table/graph with the predicted values of surface and bulk diffusion? This table could perhaps be the same table as for the model-predicted surface potential I requested above.

Page 17 lines 358-359 : “Adsorption of diffusing molecules on the channel walls would indeed lead to hindered transport. Plecis et al.²³ mention a slightly reduced diffusivity of negatively charged fluorescein that they attribute to electrostatic adsorption: in fact, it only happens when the channel walls are positively charged.” This is not correct. Plecis et al. observe a reduced diffusivity of the positively charged Rhodamine 6G which they attribute to its adsorption to the negatively charged silica walls. Such a mechanism could also be at play here for the positively charged molecules.

Reviewer #3 (Remarks to the Author):

October 10, 2017

Stephen Levy
Associate Professor
Physics Department
Binghamton University, SUNY

Review (revision): Stepping down to the ultra-nanoscale: unexpected behaviors in molecular transport through size-controlled nanochannels

The authors have done a significant amount of work improving the manuscript. They present interesting experimental results that may not have been observed before. However, I find the (mostly new) physical model that they use to explain some of their results to be unconvincing. The authors seem to want two contradictory things: to claim that they observe novel results at the ‘ultra’ nanoscale and also to claim that it is too difficult to find a theory that will take into account all the meaningful effects at the ultra nanoscale. I would have preferred if they focused on the interesting experimental results (some of which have been removed as I will discuss in a following paragraph) and removed the theory/model.

I do not find the so called ‘electro-steric’ model that they propose convincing. However, I do think that most readers can easily come to their own conclusions regarding the merits of the proposed model (which the authors do often qualify in the manuscript as not being able to describe many features of the data they observe). Therefore, I have no objection to publishing once the authors address the following concerns.

I am concerned that the authors have chosen to remove data that was previously presented in figures in the first draft. This choice appears to be motivated by the fact that they do not understand that data particularly well (I am mostly referring to Figures 3 and 4 of the first draft). The data concerned the effect of diffusion at low and high ionic strength. The ionic strength is clearly a very important experimental knob that can be used to determine the applicability of the ‘electro-steric’ model that the authors propose. I would prefer that they add these results back to the current draft, at least in the supplemental material.

I am confused by the bulk diffusivity measurements that are presented in Table 1. The authors state in the Table caption (but not that I could find directly in the text) that the bulk diffusivity row corresponds to the diffusivity as measured in 250 nm deep slits. However, it is known that diffusivity should scale inversely with particle radius (for Stokes flow at low Reynolds number as this certainly is). The authors observe that the ‘bulk’ diffusivity of histamine and epinephrine varies by an order of magnitude but at least their volume only varies by a factor of 1.7 (I could not find where the authors list the ‘radius’ of their molecules though it appears to play a primary role in the electro-steric effect as given by the limits of the integral in Eqn 2). There does not appear to be a strong correlation between the measured bulk diffusivity values and the cubed root of the volume of each molecule as listed in the Table. This should be explained.

The main result for the effective diffusivity as shown in Figure 3 is problematic. The authors rely heavily on a previous model, termed ‘enrichment’ in the manuscript, to describe the diffusivity. The model does a good job for negative molecules and for positive molecules above ~ 40 nm. Though the derivation presented in the supplementary material is complicated, the authors essentially seem to modify the enrichment theory by changing the limits of integration as given in Eqn 2 to account for the finite size of the solute molecule.

I do not understand how this leads to the non-monotonic behavior that the authors plot in Figure 3 for positive solute molecules. Is this trivially because the authors normalize by the total height of the channel ($1/h$ term in front of the integral) but are only integrating from r_s to $h-r_s$? I tried to build a quick analytic model of this using the expression for the potential as given by Eqn 18 of Plecis (1) (authors of the enrichment model) and was unable to observe the non-monotonic behavior that the authors plot.

Further, while the new electro-steric model putatively allows for the effective diffusivity to decrease for small heights, it does not do a good job of matching the data. And it appears to un-physically predict that the effective diffusion approaches zero at non-zero channel heights. And as the authors note, when the steric effect is added to the model it does an obviously worse job at describing the data for negatively charged solute molecules. Do the authors find that the benefits of including the steric effect outweigh these obvious deficiencies?

I continue to believe that at least some of the effects that the authors observe, particularly the sharp decrease in diffusivity at small channel height, is due to adsorption. The authors seem unwilling to face this possibility though it was observed in the Plecis (1) paper at even larger slit heights than investigated here. Clearly adsorption (whether or not the solute molecules have a large polarizability) plays some role here.

I am also not convinced by the authors claim that neutral molecules ‘acquire’ an effective charge by asymmetrically displacing more positive counter-ions for the small slit heights. Because of this, I find Figure 4 to be irrelevant to the manuscript.

It is fairly clear from Fig S3 and from Fig 2 that the cumulative transport for epinephrine ($q = +1$) appears more similar to aminosalicylic acid ($q = -1$) than to histamine ($q = +2$) for the channels larger than 5 nm. However this difference appears to vanish when the effective diffusion is plotted in Fig 3. This seems peculiar.

The authors should explicitly state the Debye length of the solution and probably state in the main text (rather than just in the Figure 2 caption) that the ionic concentration of the experiments is 50 mM NaCl.

I find the effective diffusivity model described in Section 12 of the supplementary information to be problematic. This is mostly because the authors use an equilibrium theory (Eqn 33) to describe a transport phenomenon that is non-equilibrium. It is also not

clear why the diffusion of the charged solute molecules through the nanoslit region will not modify the ionic concentration in that region and hence the potential. The authors actually state in Section 12 that “This is clearly not correct, and we will discuss possible consequences of this assumption later.” I failed to find any subsequent portion of the manuscript where this is discussed.

Lines 164 and 165 refer to aspirin (Fig 2c) and phenylalanine (Supplementary Information Section 7) but these labels are reversed. That is, aspirin is discussed in the supplementary information and phenylalanine is shown in Fig 2c.

References

1. Plecis, A., Schoch, R. B. & Renaud, P. Ionic Transport Phenomena in Nanofluidics:

Experimental and Theoretical Study of the Exclusion-Enrichment Effect on a Chip.

Nano Letters **5**, 1147–1155 (2005).

Reviewer #1:

Comment 1: The authors have convinced me that adsorption is not an issue for the diffusion of neutral molecules through ultra-fine nanopores, based on the polarizability values now listed in the table. I suspect then the effective net charge they see with these neutral molecules in ultra-fine nanopores is because of the pH in these pores, which is much lower than in the bulk. This would explain why aspirin seems to have a positive charge. Proteins like phenylalanine has two pKa's and it could well be that they become negative at low pH due to dissociation of certain functional groups. I would like the authors to comment on this possibility. I believe that they have found a new phenomenon but, because they do try to develop a model to explain it, all possible mechanisms should be at least discussed in the published paper. I can recommend publication if the authors can describe this pH effect in the manuscript.

Reply 1: We thank the reviewer for believing in the novelty of our results, and for agreeing that it is very hard to find explanations that stand up to scrutiny.

Concerning the possible role of the pH of the solution inside the channels: first of all, one should notice that our model accounts for the change of pH inside the nanochannel as the height h changes. This is done within the calculation of the silicon surface charge density. Two competing phenomena are balanced: the increasing attraction of positive ions into the channel due to the negatively charged silicon surfaces as the surface-to-volume ratio increases; the decrease in surface charge density of the silicon surfaces due to the decrease of the solution pH. Both effects are iteratively adjusted until convergence to a stationary state is achieved. As the reviewer remarks, we do find that the pH inside the nanochannel is lower than the one in the bulk. More precisely, we find that the positive ions concentration increases by a factor of up to 5 for the smallest nanochannels with $h=2.5$ nm. This implies a change in pH from 7.0 to 6.2. We have now better stressed this point in the manuscript.

Concerning the role of such changes in pH for aspirin and phenylalanine: it is known, and we have checked using the online computational tools of chemicalize.org, that the hydroxyl group of aspirin has a pKa of 3.5; at the pH that we estimate inside the channels (< 3) the OH group dissociates very little, and the fraction of molecules with non-dissociated OH groups increases towards 100%, so that the solute is, as we state, mostly neutral (see Fig. 1 below). Also, aspirin is hydrophobic, as we mention in the text (see table I in the paper). It seems safe to conclude that in no way aspirin can acquire a positive charge in our solutions, whatever the pH.

The situation is also quite complicated with phenylalanine that, as the reviewer correctly remarks, has two pKas, because of the simultaneous presence of a hydroxyl and an amine group. At very acidic pH, the OH group does not dissociate, while the NH_2 protonates, so that the molecule has a positive charge. At pH around 6, the solute is mostly neutral, and only acquires a negative charge at $\text{pH} > 9$, when the OH group dissociates (see Fig. 2 below). Again, calculations support our presumption that phenylalanine should not exhibit a negatively charged molecule behavior in our experimental conditions. As suggested by the reviewer, we have added a short discussion to our text, and more details in the SI.

Isoelectric point

Figure 1. Microspecies % distribution (above) and charge (below) for aspirin as a function of pH.

Isoelectric point

Isoelectric point: 5.96

Figure 2. Microspecies % distribution (above) and charge (below) for phenylalanine as a function of pH.

Reviewer #2:

Comment 1: In general the paper uses many hyperbolic words: “unprecedented”, “first of its kind”, “unique”. As I will show below, experiments like this have been extensively performed in the 1980s and have been reported in a number of papers. In these experiments generally track-etched membranes were used, and the influence of diffusing molecular size and pore size were experimentally studied and theoretically analyzed by a sophisticated model including hindered diffusion on top of steric hindrance and electrical charge (the authors only consider the last two). The only aspect in which the present paper is truly new is in the use of a new geometry (slit-type pores) and probably a better controlled pore homogeneity. That however is all, and theoretically it still lags the 1980 papers. For example, it does not reach the quality and thoroughness of the paper of Deen and Smith “Hindered diffusion of synthetic polyelectrolytes in charged microporous membranes”, J. Membrane Sci. 12 (1912) 217-237. With respect to the previous paper the paper has much improved, and some fundamental issues have been clarified with respect to the previous version, especially by employing the theory from the Plecis paper. However, when reading the revised version, the reviewer realized that up till now a centrally important aspect of the transport has been neglected: hydrodynamic interactions have been neglected, while static interactions with the wall were accounted for (electrical forces, Born repulsion and van der Waals forces). The reviewer apologizes he did not notice this before. A central concept in hydrodynamics is that of hindered diffusion by hydrodynamic interaction of the diffusing particle with the wall. In the 1987 review paper of Deen (to which the authors refer), they will find both experimental and theoretical information on the hindered diffusion coefficient, in this paper denoted by K_d . Most interestingly for the present paper, figures 6 and 7 in the Deen paper show experimental data on hindered diffusion of small solutes in track-etched pore membranes with pore radii of 4.5 – 30 nm, in the same range as the present paper. The present paper now needs an added analysis of its data in light of these historical papers that investigate systems of the same size (pore diameter), though different geometry (cylindrical instead of slit-type). The historical papers concerned are mentioned in the Deen paper on pages 1418 and 1419. Especially the paper of Deen and Smith from 1982 performs a thorough analysis of a nanopore system, in which electrostatic effects, steric effects and hindered diffusion are all considered, and the surface charge is even independently measured using streaming current measurements. Notably in these papers, a strongly decreased diffusional transport is observed already when the particle radius is 0.2 times the pore radius. **In light of these works I conclude that it is imperative that the authors include hindered diffusion in their model.** The values of D/D_0 for slit-type pores can e.g. be extracted from eqn 32 in the Deen paper.

Reply 1: We acknowledge that the reviewer consider that the revised version has “much improved” our paper, and we are grateful for the reviewer’s contribution to this improvement. We agree with the reviewer’s comment, that one aspect in which the present paper is new is in the use of a new geometry (slit-type pores) and “probably a better controlled pore homogeneity”,

Fig. 3. Pore size distribution for Membrane 48.

Biochim. Biophys. Acta, 255 (1972) 273-303

Figure 3. Typical pore size distribution reported by Beck and Schultz (1972).

although we would certainly omit the “probably”. We have stressed several times that the “unprecedented” experiments we perform are only possible because of the “unique” characteristics of our membranes and would have not been possible with track etched membranes, for which dimensional tolerances are known to be wide (see Figure 3). Same is true for other membrane types including carbon nanotubes, TiO₂, among others. However, we do believe that this is not the unique novelty of our study. In fact, we investigate the diffusive transport at a scale that was not achievable before. The reviewer

consider the work of Deen with “track- etched pore membranes with pore radii of 4.5 – 30 nm”, to be “in the same range as the present paper”. We respectfully disagree. As we studied the

Table I. Dimension of nanochannels studied in historical manuscripts on the topic in the literature.

Authors	Year	Diameter (nm)
Smith and Deen	1980	20
Smith and Deen	1982	30-44
Smith and Deen	1987	40-100
Beck and Schultz	1972	9-64
Mitchell and Deen	1986	40
Van Bruggen et al.	1974	30
Bohrer, Patterson, and Carroll	1983	12-150
Cannell and Rondelez	1980	54-258

historical papers on the topic, we observed than the work by Deen was in fact limited by the use of membranes with channels of 20 nm in diameter (see Table I). The smallest channel investigated were reported by Beck and Schultz in 1972 with a diameter > than 9 nm. As such, these study could not capture the complexity of fluids at what we call the ultra-

nanoscale, which in contrast, is the focus of our work. We may just add that our results, as all reviewers explicitly acknowledge, are also new, and unexpectedly so.

The reviewer correctly claims that the novelty does not extend to the models we use. Indeed, we took great care to adapt established models to our situation. However, a further improvement to the modeling part was nonetheless required, according to the reviewer: the addition of hydrodynamic interactions and the resulting hindered diffusion effects. We thank the reviewer for pointing out this oversight on our part. By adding these effects in the model we obtained a state of the art sophisticated model including hydrodynamic interactions on top of steric hindrance and electrical charge. Notably and expectedly we observed that accounting for hindrance effects did not improve the model ability to capture our experimental results. On the contrary, it clearly shows to the community that our results sit outside the realm of what has been observed and modeled to date.

The reviewer should note that in the previous version of the manuscript we were attempting to obtain the best fit between models and experimental data. As such, we used both molecular

charge and molecular radius as free parameters. For this current version of the paper, molecular radius and charge were selected based on the actual molecular properties determined through chemicalized.org. We no longer attempt to fit the data. Therefore, it should not be surprising that the model's ability to match the experimental data is further reduced.

We added a paragraph arguing that hydrostatic effects cannot play an important role in our systems. It can be noted that the molecules studied by Smith and Deen in their 1982 paper, have molecular weights 2 to 3 order of magnitude larger than our molecules, with diameters which are more than 10 times larger. If hydrodynamic interactions can be surmised to be important for molecules that are much bigger than the solvent, the same is not necessarily true for our molecules, whose effective diameters span just 3 to 5 water molecules. Let us assume, for the sake of the argument, that hydrodynamic interactions are important, and that their effect can be correctly described by expressions derived for spherical particles (although our molecules are very far from being spherical); they should be most relevant for neutral molecules, which by definition should be insensitive to electrostatics. We should then find that the effective diffusivity of aspirin or phenylalanine slowly decreases monotonically from the bulk value, until steric effects kick in, and the diffusivity vanishes. Instead, we observe the puzzling, and in the case of aspirin non monotonic, behavior that we discuss in the paper. We have made this point quantitative for neutral molecules, and the effective diffusivity predicted by a complete model including hard-sphere, as well as hydrodynamic interactions, is now shown in the figures for aspirin and phenylalanine.

Separate issues:

Comment 2: I do not understand why the charge of aspirin has such a large influence on the predicted transport rate. The experiments with aspirin have been performed at pH = 3, where the silica is practically uncharged, so that electrostatic effects should be negligible.

Reply 2. We agree with the reviewer comment that aspirin does not show the predicted diffusivity typical of a neutral molecule in an uncharged channel, however we assume that this peculiar behavior may be due to the molecule hydrophobicity.

Comment 3. In the context of the above remark, can the authors in the supplementary information section 10 provide a table with the model-predicted surface potential at the different pH values used?

Reply 3. We have performed additional measurements of the surface potential with the method of the streaming potential. A figure was added to the supplementary information, describing the change in surface potential as a function of the solution pH. Figure 4 in the main text has been added: it shows the surface potential measured as a function of the nanochannel heights, and compares it to the prediction of the model.

Comment 4. The model presented in section 10 of the supplementary information seems to assume that the diffuse layer potential ψ_d is located at $z=h$. This implies that h denotes half the channel height, while it denotes the full channel height in the rest of the paper. This should be clarified. It also possibly has led to a mistake with a factor of two.

Reply 4. The diffuse layer potential is located at the opposite surfaces at $z=0$ and $z=h$. A figure describing the reference system has been added in the Supplementary Information Section 10.

Comment 5. The effective charge argument of the authors (supplementary information 13) is impossible to follow. In my opinion the underlying physics is erroneous. As it is now, it severely undermines the believability of the entire paper, as it (at least for me) shows a rather casual attitude to theory development. In the generally used model, the z -dependent anion and cation concentration in the channel is purely determined by the Boltzmann distribution between bulk and channel, based on differences in the local (electrical, van der Waals,...) potential.

Reply 5. The generally used model computes the diffusivity of a charged solute assuming that its charge couples to the electrostatic potential inside the channel, which depends only on the equilibrium properties of the solvent. In other words, the presence of the charged solute does not alter the distribution of the potential inside the channel. This is clearly wrong, and our effective charge argument was an attempt to address this drawback of the standard model. We agree that its formulation was rather confused and confusing, and we have decided to suppress it altogether.

Comment 6. Concentrations of other species, when equal to their bulk concentration, do not figure as they do not cause changes in the interaction potential. An argument on the presence of other molecules can only be constructed when they influence the local energy state (potential) of the molecule under consideration at the walls to a different extent than the local energy state in the bulk. That can be the case for example when they adsorb to large extent and change the local environment at the wall. In that case however other equations should be used than the ones now used and a great number of assumptions and additional measurements will be needed to prove the hypothesis.

Reply 6. We are thankful for the remark.

Small points: Comment 7: Page 2 lines 42/43: “Importantly, experiments distinctly showed that the diffusive transport of cationic, anionic, as well as neutral species at the ultra-nanoscale is dominated by electro-steric effects and charge density.” “
” is an unknown term that needs definition, for example as “electrical and steric”. The “...and charge density” raises the question: charge density of what: of the diffusing molecule or the pore wall? If so, how is this different from the “electro-“ part of the “electro-steric” effect? If it just adds a further qualification, the sentence should reflect this.

Reply 7: We appreciate the reviewer’s comment and clarified the terminology and sentence in

the manuscript.

Comment 8: Page 3 lines 66/67: “In this regime, all molecules appear to possess an electric charge, either actual or effective, so that transport of all types of solute is determined by the interplay of electrostatic, steric and volume-related effects.”: what is the difference between steric and volume-related effects?

Reply 8: The sentence has been reworded to avoid confusion.

Comment 9: Table 1: are the bulk diffusion coefficients comparable to the literature-reported values for bulk diffusion?

Reply 9: We apologize with the readers for a misuse of terminology regarding “*bulk diffusion coefficient*”, and we thank the reviewer for pointing out the inconsistency. Extracting the effective diffusivities inside the nanochannels using Eq. (1) requires knowledge of the effective diffusivities inside the microchannels which, similarly to bulk diffusivities, are not available in the literature for these molecules. Since these values are unknown, we take the effective diffusivities in a 1/4 micron channel (250 nm) as plausible estimates of the diffusivities in a 1 micron channel (the dimension of the microchannels in the membrane). We stress that these effective diffusivities are not expected to compare quantitatively with values of single-molecule diffusion coefficients that one may compute from the Einstein-Stokes relation.

Comment 10: Page 7: At 50 mM NaCl the Debye length is about 2 nm. I am surprised that the EEM produce the predicted dominance of surface diffusion even at 40 nm, when $h/L_D = 20$... Can the authors provide a table/graph with the predicted values of surface and bulk diffusion? This table could perhaps be the same table as for the model-predicted surface potential I requested above.

Reply 10: By calculating the potential distribution between the silicon surfaces we were able to assess that the 40 nm nanochannels have a considerable region (approximately 50%) greatly affected by the EDL. Only the solution within the largest nanochannels (250 nm) can be safely assumed to be unaffected by the surface charges. Additionally, we would like to clarify that the tabulated bulk diffusion coefficient for each molecule corresponds to the diffusivity extrapolated from the 250 nm nanochannels results. The plotted diffusivities are “effective” values obtained from the experimental data. We did not attempt to predict or separate bulk versus surface diffusivities at different nanochannel sizes. This would require substantial assumptions, which would limit the relevance of data.

Comment 11: Page 17 lines 358-359 : “Adsorption of diffusing molecules on the channel walls would indeed lead to hindered transport. Plecis et al.²³ mention a slightly reduced diffusivity of negatively charged fluorescein that they attribute to electrostatic adsorption: in fact, it only happens when the channel walls are positively charged.” This is not correct. Plecis et al. observe

a reduced diffusivity of the *positively charged* Rhodamine 6G which they attribute to its adsorption to the negatively charged silica walls. Such a mechanism could also be at play here for the positively charged molecules.

Reply 11: The sentence is modified for clarity. We add that adsorption may occur in our system, although it does not provide an explanation for the results obtained with the various molecules.

Reviewer #3:

Review (revision): Stepping down to the ultra-nanoscale: unexpected behaviors in molecular transport through size-controlled nanochannels

Comment 1: The authors have done a significant amount of work improving the manuscript. They present interesting experimental results that may not have been observed before. However, I find the (mostly new) physical model that they use to explain some of their results to be unconvincing. The authors seem to want two contradictory things: to claim that they observe novel results at the ‘ultra’ nanoscale and also to claim that it is too difficult to find a theory that will take into account all the meaningful effects at the ultra nanoscale. I would have preferred if they focused on the interesting experimental results (some of which have been removed as I will discuss in a following paragraph) and removed the theory/model. I do not find the so called ‘electro-steric’ model that they propose convincing. However, I do think that most readers can easily come to their own conclusions regarding the merits of the proposed model (which the authors do often qualify in the manuscript as not being able to describe many features of the data they observe). Therefore, I have no objection to publishing once the authors address the following concerns.

Reply 1: We appreciate the reviewer comment regarding the improvements in the work as well as the novelty of the results. We compared our data to the state of the art theoretical understandings of molecular diffusion in nanopores. The model now includes hydrodynamic interactions on top of steric hindrance and electrostatics as also strongly suggested by reviewer 2. We perhaps made an unfortunate choice of name for the model (electro-steric). The model is not new and follows the work by Smith and Deen in the 1980’s, and it is the same as the one quoted by the second reviewer. We have only made explicit the calculation of the electrostatic potential inside the channel, with a boundary charge that is self-consistent computed with an equilibrium protonation-deprotonation model of the electric charges on the silica surface in line with what done by *Plečis et al.* The main idea behind this theoretical exercise was to verify if the current understanding of diffusive transport in nanoconfinement could explain the results. Notably we observed that this comprehensive model does not capture neither qualitatively nor quantitatively our experimental results. On the contrary, it clearly shows to the community that our results sit outside the realm of what has been observed and modeled to date.

The reviewer should note that in the previous version of the manuscript we were attempting to

obtain the best fit between models and experimental data. As such, we used both molecular charge and molecular radius as free parameters. For this current version of the paper, molecular radius and charge were selected based on the actual molecular properties determined through chemicalized.org. We no longer attempt to fit the data. Therefore, it should not be surprising that the model's ability to match the experimental data is further reduced.

In the previous version of the article, we expanded the theory in the attempt to explain what we called "the effective charge hypothesis". However, after thoughtful consideration and the reviewer's comment, we decided to keep the development of a novel theoretical analysis for future studies. At this time, we will limit ourselves in presenting only our experimental data compared with the currently available mathematical model.

Comment 2: I am concerned that the authors have chosen to remove data that was previously presented in figures in the first draft. This choice appears to be motivated by the fact that they do not understand that data particularly well (I am mostly referring to Figures 3 and 4 of the first draft). The data concerned the effect of diffusion at low and high ionic strength. The ionic strength is clearly a very important experimental knob that can be used to determine the applicability of the 'electro-steric' model that the authors propose. I would prefer that they add these results back to the current draft, at least in the supplemental material.

Reply 2: The data set in Figure 3 of the first draft has been reanalyzed and broadly expanded and it is now included as Figure 5. The results are now presented as the ratio between the effective diffusivities obtained at high versus low ionic strength for histamine and cefazolin at each nanochannel size. The results provide a solid demonstration that, while exclusion and enrichment effects play a role in the transport of negative or positive charges in nanochannels, they cannot explain the drop in diffusivity observed at the ultra-nanoscale.

After thoughtful consideration and the reviewer's comment, we decided to keep the development of a novel theoretical analysis for future studies. As such, that part of the manuscript including figure 4 have now been removed.

Comment 3: I am confused by the bulk diffusivity measurements that are presented in Table 1. The authors state in the Table caption (but not that I could find directly in the text) that the bulk diffusivity row corresponds to the diffusivity as measured in 250 nm deep slits. However, it is known that diffusivity should scale inversely with particle radius (for Stokes flow at low Reynolds number as this certainly is). The authors observe that the 'bulk' diffusivity of histamine and epinephrine varies by an order of magnitude but at least their volume only varies by a factor of 1.7 (I could not find where the authors list the 'radius' of their molecules though it appears to play a primary role in the electro-steric effect as given by the limits of the integral in Eqn 2). There does not appear to be a strong correlation between the measured bulk diffusivity values and the cubed root of the volume of each molecule as listed in the Table. This should be explained.

Reply 3: We would like to clarify that the tabulated diffusivities for each molecule corresponds to the diffusivity extrapolated from the 250 nm nanochannels results. Extracting the effective diffusivities inside the nanochannels using Eq. (1) requires knowledge of the effective diffusivities inside the microchannels which, similarly to bulk diffusivities, are not available in the literature for these molecules. Since these values are unknown, we take the effective diffusivities in a 1/4 micron channel (250 nm) as plausible estimates of the diffusivities in a 1 micron channel (the dimension of the microchannels in the membrane). We stress that these effective diffusivities are not expected to compare quantitatively with values of single-molecule diffusion coefficients that one may compute from the Einstein-Stokes relation.

Comment 4: The main result for the effective diffusivity as shown in Figure 3 is problematic. The authors rely heavily on a previous model, termed ‘enrichment’ in the manuscript, to describe the diffusivity. The model does a good job for negative molecules and for positive molecules above ~40 nm. Though the derivation presented in the supplementary material is complicated, the authors essentially seem to modify the enrichment theory by changing the limits of integration as given in Eqn 2 to account for the finite size of the solute molecule.

I do not understand how this leads to the non-monotonic behavior that the authors plot in Figure 3 for positive solute molecules. Is this trivially because the authors normalize by the total height of the channel ($1/h$ term in front of the integral) but are only integrating from r_s to $h-r_s$? I tried to build a quick analytic model of this using the expression for the potential as given by Eqn 18 of Plecis (1) (authors of the enrichment model) and was unable to observe the non-monotonic behavior that the authors plot.

Reply 4: The hard-sphere potential prevents the solute molecule from approaching the walls closer than their radius, so that a region of size $2r_s$ is excluded from diffusion. The diffusivity then vanishes at $h = 2r_s$. This is indeed the origin of the non-monotonic behavior of the diffusivity, as the reviewer correctly points out.

Comment 5: Further, while the new electro-steric model putatively allows for the effective diffusivity to decrease for small heights, it does not do a good job of matching the data. And it appears to un-physically predict that the effective diffusion approaches zero at non-zero channel heights. And as the authors note, when the steric effect is added to the model it does an obviously worse job at describing the data for negatively charged solute molecules. Do the authors find that the benefits of including the steric effect outweigh these obvious deficiencies?

Reply 5: As we explained above, we compared our data with the current theoretical understanding of transport in nanoconfinement with the purpose of testing if our surprising results could be explained by what is already known. One of the puzzling aspects of our results is precisely that the more refined the model—namely with the addition of steric (hard-sphere) and hydrodynamic interaction effects—the worse the agreement with the experimental data.

Physically, one would think (as the second reviewer also does) that all these interactions with the channel walls should be accounted for. It seems only fair to conclude that including these effects in the model as it is usually done, does not do a good job in describing the data. This supports our conclusion that our data are in fact novel.

Comment 6: I continue to believe that at least some of the effects that the authors observe, particularly the sharp decrease in diffusivity at small channel height, is due to adsorption. The authors seem unwilling to face this possibility though it was observed in the *Precis* (1) paper at even larger slit heights than investigated here. Clearly adsorption (whether or not the solute molecules have a large polarizability) plays some role here.

Reply 6: We note that even the various reviewers appear to have conflicting views on adsorption. It certainly is another form of interaction with the channel walls that should be included in the models and it is a challenge to be addressed in future theoretical work. While we agree that adsorption may be occurring in our channels and we indicate this in the manuscript, we also feel that adsorption itself is not able to explain the results. Further, any discussion of adsorption must be quantitative. As happy as we would be to be able to perform such quantitative assessment, we are not, and we prefer not to introduce speculations that can be left to the readers.

Comment 7: I am also not convinced by the authors claim that neutral molecules ‘acquire’ an effective charge by asymmetrically displacing more positive counter-ions for the small slit heights. Because of this, I find Figure 4 to be irrelevant to the manuscript.

Reply 7: Figure 4 was removed as well as the effective charge hypothesis.

Comment 8: It is fairly clear from Fig S3 and from Fig 2 that the cumulative transport for epinephrine ($q = +1$) appears more similar to aminosalicic acid ($q = -1$) than to histamine ($q = +2$) for the channels larger than 5 nm. However this difference appears to vanish when the effective diffusion is plotted in Fig 3. This seems peculiar.

Reply 8: We acknowledge that the difference in timescale in the cumulative release graph may have generated confusion. For convenience we plot here the same graphs but highlighting the first part of the release for histamine (see Figure 4). Rather than comparing the graphs in terms of the linearity of curves, one should note that for both epinephrine and histamine, three groups of curves can be observed: I) overlapping cumulative release profiles (13, 20, and 40 nm channels for epinephrine; 5.7, 13, 20, 40 nm for histamine); II) distinct and higher release profiles for 250 nm membranes; III) distinct and lower release profiles for the smaller nanochannel sizes. The overlap between curves in the ~5.7-40 nm range is what we attribute to a near-surface diffusion for the positive charges. In the case of aminosalicic acid the cumulative release curves for each nanochannel size are distinct and not overlapping. This is what we ascribe to the gated transport

for negatively charged molecules.

Figure 4. Cumulative release profiles (above) and release rates (below) for 3-Aminosalicylic acid, epinephrine, and histamine.

Comment 9: The authors should explicitly state the Debye length of the solution and probably state in the main text (rather than just in the Figure 2 caption) that the ionic concentration of the experiments is 50 mM NaCl.

Reply 9: This is now done.

Comment 10: I find the effective diffusivity model described in Section 12 of the supplementary information to be problematic. This is mostly because the authors use an equilibrium theory (Eqn 33) to describe a transport phenomenon that is non-equilibrium. It is also not clear why the diffusion of the charged solute molecules through the nanoslit region will not modify the ionic concentration in that region and hence the potential. The authors actually state in Section 12 that “This is clearly not correct, and we will discuss possible consequences of this assumption later.” I failed to find any subsequent portion of the manuscript where this is discussed.

Reply 10: The model is problematic: we agree with the reviewer, and we stress that this is the same remark as we made at the beginning. Equation 33 (of the old SI) describes the equilibrium distribution of ions in the nanochannels. When solute molecules enter the channel under a concentration gradient, they should, in principle, change the local charge distribution, which should in turn affect their diffusivity. This phenomenon is neglected in all published models that instead treat the solute as a “weak perturbation” to the ion distribution—and thus to the electrostatic potential—inside the channel. Since this perturbation is weak, it can be ignored. The whole process is indeed quite unsatisfactory, because no proof is given of the “weakness” of such a perturbation, nor of the absence of serious consequences in ignoring it. However, it is also

clear that including such a perturbation is not obvious, although we suspect it might be related to our puzzling experimental observations; we have tried to present a semi-quantitative argument supporting this suspicion, but we feel now that they are very premature, and must be much more carefully crafted before publication.

Comment 11: Lines 164 and 165 refer to aspirin (Fig 2c) and phenylalanine (Supplementary Information Section 7) but these labels are reversed. That is, aspirin is discussed in the supplementary information and phenylalanine is shown in Fig 2c.

Reply 11: The mistake was corrected.

REVIEWERS' COMMENTS:

Reviewer #1 (Remarks to the Author):

The authors have provided more data to confirm that adsorption and pH change are not responsible for the strange phenomenon they observed--that neutral molecules exhibit unusual diffusive transport in a nanochannel. The only explanation seems to be hydrophobicity. Although more experiments can be carried out to test this hypothesis, I think the paper has demonstrated enough novelty and careful experimentation to warrant publication.

I hence recommend publication in Nature Communications in the present form.

Reviewer #3 (Remarks to the Author):

My overall impression of the manuscript is largely unchanged: the authors have made interesting experimental observations and spend an excessively large amount of space in the manuscript discussing a theoretical model that is at best ad hoc.

The manuscript is difficult to follow and in many places I cannot say with confidence that I understand how calculations were made.

I still don't understand how the authors can ignore the fact that the extrapolated diffusivity in a very deep nanoslit (250 nm) doesn't correlate with any known characteristics of the molecules they study. They seem to argue that we shouldn't expect the bulk diffusivity measurement to be explainable since their molecules are 'complicated.' Fine, but then how is the reader then to proceed to understand the diffusivity measurements in smaller nanoslits?

Can the authors explain why some data points in Fig 3 (like in Fig 3c) have much larger uncertainty than other points at the same slit depth?

Can the authors estimate the uncertainty on the diffusion shown to 3 digits in Table 1?

Can the authors comment on how their site-binding model parameters (surface density, equilibrium constant, and double layer capacitance) compare to others in the literature?

Are the theory and experimental curve shown in Fig 4 independent? It appears from the supplemental information that the zeta potential is related to the surface potential using Debye-Huckel theory. Does this mean the curves are correlated?

Page 13 states 'see details in Supplementary Information Section 10.' This doesn't appear to be the correct section.

Is an equation missing from page 14? The text states ‘the neutral partition coefficient should then read.’ It seems like something should follow this clause.

I don’t really understand why the authors write the Ganatos drag coefficient as a function of two variables when they only ever compute it as a function of one variable (r/h).

I don’t understand the discussion following Figure 5. It seems to contradict data shown in the figure.

The method the authors use to compute the potential in the channel is no longer clear to me. They have removed several equations from the supporting information that previously described the process. In fact, the SI now states in Section 13 ‘computed from Eq. (??) yields the Stern layer potential with the use of Eq. (??)’. This needs to be fixed.

Replies to the reviewer's comments:

Comment 1. My overall impression of the manuscript is largely unchanged: the authors have made interesting experimental observations and spend an excessively large amount of space in the manuscript discussing a theoretical model that is at best ad hoc.

Reply 1. The model included in the manuscript represents the state of the art understanding of diffusive transport in confined spaces. The model description is complex and requires a sufficient amount of space to be described. Its role in the manuscript is to demonstrate that the current understanding of the transport of molecules in small nanochannel is quite incomplete and the experimental results obtained in this study are novel.

Comment 2. The manuscript is difficult to follow and in many places I cannot say with confidence that I understand how calculations were made.

Reply 2. The result section of the manuscript has been split in subheadings to improve the clarity of presentation. The Supplementary Notes have also been arranged according to the editorial office to improve clarity.

Comment 3. I still don't understand how the authors can ignore the fact that the extrapolated diffusivity in a very deep nanoslit (250 nm) doesn't correlate with any known characteristics of the molecules they study. They seem to argue that we shouldn't expect the bulk diffusivity measurement to be explainable since their molecules are 'complicated.' Fine, but then how is the reader then to proceed to understand the diffusivity measurements in smaller nanoslits?

Reply 3. Measurements of diffusion coefficient of molecules in the bulk are affected by substantial variability. Large variability in diffusivity is found in the literature even for well know and thoroughly studied molecules such a glucose. Bulk diffusivity values for most molecules in our studies are not available in the literature. Similarly, values are not available for diffusion in microchannels. Since these values are unknown, we take the effective diffusivities in a 1/4 micron channel (250 nm) as plausible estimates of the diffusivities in a 1 micron channel (the dimension of the microchannels in the membrane). We stress that these effective diffusivities are not expected to compare quantitatively with values of single-molecule diffusion coefficients that one may compute from the Einstein-Stokes relation. Rather than emphasizing the absolute diffusivity values, in this work we focus on the relative changes in diffusivity at different size of channels and different molecular charge.

Comment 4. Can the authors explain why some data points in Fig 3 (like in Fig 3c) have much larger uncertainty than other points at the same slit depth?

Reply 4. Larger uncertainty in results can be ascribed to experimental variability.

Comment 5. Can the authors estimate the uncertainty on the diffusion shown to 3 digits in Table 1? **Reply 5.** In light of the reviewer's comment we have included the uncertainty in diffusivity in Table 1.

Comment 6. Can the authors comment on how their site-binding model parameters (surface density, equilibrium constant, and double layer capacitance) compare to others in the literature?

Reply 6. Our parameters for the site-binding model are tightly correlated to others found in the literature such as: Taghipoor et al. (2012), Behrens and Grier (2001), Siria et al (2013), and Hiemstra et al (1989).

Comment 7. Are the theory and experimental curve shown in Fig 4 independent? It appears from the supplemental information that the zeta potential is related to the surface potential using Debye-Huckel theory. Does this mean the curves are correlated?

Reply 7. The experimental result and the theory in figure 4 are independent. The experimental results provide a semi-quantitative validation of the site-binding model in its ability to describe the variation of surface potential with a variation of the nanochannel size.

Comment 8. Page 13 states 'see details in Supplementary Information Section 10.' This doesn't appear to be the correct section.

Reply 8. The reference to the section in the Supplementary Information has been corrected.

Comment 9. Is an equation missing from page 14? The text states 'the neutral partition coefficient should then read.' It seems like something should follow this clause.

Reply 9. We thank the reviewer for pointing out this mistake. The missing formula related to the partition coefficient has been added.

Comment 10. I don't really understand why the authors write the Ganatos drag coefficient as a function of two variables when they only ever compute it as a function of one variable (r/h).

Reply 10. There is no available equation that fully describes the drag coefficient for non-cylindrical geometries. However, the drag coefficient at the channel centerline is commonly used in the literature as a first approximation. This approach has been adopted in several works including Anderson & Quinn (1974), Bungay & Brenner (1973), Brenner & Gaydos (1977), Mavrovouniotis & Brenner (1986), and for slit nanochannels in Ganatos et al. (1980), Weinbaum (1981), Happel and Brenner (1983).

Comment 11. I don't understand the discussion following Figure 5. It seems to contradict data shown in the figure.

Reply 11. The results in figure 5 clearly demonstrate that enrichment and exclusion effects play a role in the transport of positive and negative charges, respectively. This is evidenced by the substantial change in diffusivity observed for most of the smaller channel sizes with a variation in ionic strength of bulk solution. However, it also indicates quite notably that neither enrichment nor exclusion effect can explain the abrupt drop in diffusivity observed at the ultra-nanoscale.

Comment 12. The method the authors use to compute the potential in the channel is no longer clear to me. They have removed several equations from the supporting information that previously described the process. In fact, the SI now states in Section 13 'computed from Eq. (??) yields the Stern layer potential with the use of Eq. (??)'. This needs to be fixed.

Reply 12. The Supplementary Information and related equations have been fixed.